# LOSSGATE: INCOMPLETE INFORMATION AND MISALIGNED INCENTIVES HINDER REGULATION OF SOCIETAL RISKS IN MACHINE LEARNING

## ABSTRACT

Regulators seek to curb the societal risks of machine learning; a common aim is to protect the public from excessive privacy violations or bias in models. In the status quo, regulators and companies independently evaluate societal risk. We find that discrepancies in these evaluations can be either a detriment or an advantage for companies. To abide by regulation, a company needs to conservatively evaluate risk: it should train its model such that risk remains below the acceptable threshold—even if the regulator's evaluation returns higher risk measurements. This decreases model utility (up to 8%, in our experiments). Conversely, when the regulator's measurements are consistently lower than theirs, we find that a company can behave strategically and game regulation to train more accurate models. We call this *Lossgate*, an allusion to Dieselgate in environmental regulation: Volkswagen produced cars that limited their emissions when being subjected to a regulator's emissions measurement. To model incomplete information and the misaligned incentives that explain Lossgate, we leverage game theory. We obtain SPECGAME, a model for regulator-company interactions which allows us to estimate the excessive risk that results from the strategic behavior observed in Lossgate. We show Lossgate costs 70–96% higher compared to collaborative regulation in the sum cost for all players.

## 1 INTRODUCTION

The societal risks of deploying machine learning (ML) are well documented. To contain these risks, companies are increasingly expected to deploy ML algorithms that have been adapted to support algorithmic fairness (Pedreshi et al., 2008; Calders & Verwer, 2010), privacy (Blum et al., 2005; Abadi et al., 2016), robustness (Szegedy et al., 2013), or interpretability (Linardatos et al., 2020), to name a few. Meanwhile, regulators around the world seek to enforce new legislation that captures the public's expectation of how strongly an ML application should contain the associated societal risks. For instance, a regulator may enforce a maximum limit on privacy and fairness violations.

Regulators and companies currently evaluate societal risk independently from one another. This is because the regulator is a separate entity from the company. Two key issues arise from this separation: misaligned incentives and imperfect information. Recall our running example of a regulator that wishes to enforce a maximum limit on privacy and fairness violations. In this example, the company instead wishes to maximize model accuracy—which can be interpreted as a proxy for financial profit. If we visualize the trade-offs between model accuracy and societal risks (i.e., privacy and fairness violations) realized by a given training algorithm, we obtain the Pareto frontier in Figure 1:

- *Misaligned incentives* lead the regulator and company to prefer two very different points on this Pareto frontier. The regulator prefers point (A) with minimal privacy and fairness violation. Instead, the company prefers point (B) which maximizes model accuracy.
- *Imperfect information* implies that the regulator and company work with slightly different Pareto frontiers. This is due to differences in their respective estimations of societal risks. The less transparent a company is towards the regulator, the more imperfect information is.

One of our key contributions is showing how imperfect information, when combined with misaligned incentives, can be either a detriment or an advantage for companies. If a company genuinely aims to abide by legislation, it will account for possible differences between the regulator's and the company's estimations of societal risks. This is detrimental to the company; it will train a model that is over-conservative by picking a point on the Pareto frontier that is strongly in favor of reducing the societal risks, at the cost of producing lower-accuracy models. Instead, if a company 'bends the law,' it can behave strategically and leverage any difference between the regulator's and the company's estimations of societal risks to train models whose accuracy is higher—hence increasing financial profit. Put another way, the company is adopting an anti-conservative trade-off: the point it picks on the Pareto frontier is strongly in favor of producing the most useful model at a larger societal risk. We refer to this failure mode as *Lossgate*, alluding to Dieselgate in environmental regulation (see abstract).

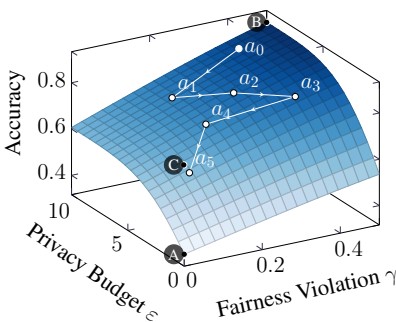

Figure 1: **Companies can behave strategically to (temporarily) achieve B but over multiple interactions $\{a_t\}$ with the regulator, this strategy, compared to collaboration C, will lead to worse outcomes for all parties due to repeated release of untrustworthy models and the resulting fines.**

These two issues, imperfect information and misaligned incentives, would not exist if the regulator and company were a single entity. This is of course not possible. Hence, we cast ML regulation as a *principal-agent problem* (PAP), the canonical framework in agency theory (Eisenhardt, 1989), commonly employed in risk analysis to formalize industrial regulation like environmental (Bier & Lin, 2013) and financial regulation (Alexander, 2006). The PAP formulation of regulator-company interactions defines a game, SPECGAME, where the regulator and company take turn in assigning penalties and releasing models, respectively.

We can then use game theory to analyze SPECGAME and design effective regulation; that is, avoid unnecessary societal risks and unnecessary economic expenditure (i.e., loss of model accuracy). To do so, effective regulation guides the regulator and company towards behavior that is closest to collaboration, as if they were making decisions as a joint committee (i.e., virtually becoming a single entity). We call this ideal setting COLLABREG and use it as a frame of reference for SPECGAME.

In the illustrative Figure 1, the outcome of COLLABREG is for the committee to choose C, while that of SPECGAME is closer to the sequence $\{a_t\}$ of interactions between the regulator and the company. Note that, although the final outcome of both is adoption of C, in SPECGAME both agents fared worse: each release of an untrustworthy model harmed the public and each penalty costed the company money. In other words, strategic behavior is inherently *inefficient*. We quantify this inefficiency by the ratio of the sum cost of regulators and the companies in SPECGAME vs. COLLABREG. We empirically find that strategic behavior collectively cost all entities involved up to 96% higher than collaboration using models trained on 6 tabular and vision datasets.

Simulating the outcomes of SPECGAME benefits both regulators and companies. For companies, we show that even in the absence of strategic behavior, imperfect information leads to excessive utility loss—by up to 8%. This is the result of uncertainty in privacy risk estimations. Hence, increased transparency from the company can in fact benefit the company itself. For regulators, our work stresses the need for regulation that is not only data-driven (Hildebrandt, 2018) and task-adapted (Coglianese, 2023) but also cognizant of the socioeconomic context of ML models. SPECGAME enables this because regulators can simulate the outcome of their policies in a *virtual regulatory sandbox* (Jeník & Duff, 2020) before deploying them. As a concrete example, we demonstrate that for a gender classification application, regulators can enforce a privacy budget $\varepsilon$ that is on average 6 lower if they initiated SPECGAME by specifying their desired guarantee first. This comes at negligible expense for the company in terms of accuracy. In summary,

- We formulate, for the first time (to the best of our knowledge), regulation of trustworthy ML as a Principle-Agent problem (PAP), the canonical framework to formalize industrial regulation. We highlight the separation between the regulator and the company and the imperfect information and misaligned incentives that ensues.
- We demonstrate that uncertainty in trustworthy auditing causes utility loss—up to 8% in the UTKFace dataset—due to incomplete information between the regulator and the company.

- To capture the risk of misaligned incentives and strategic behavior, we introduce SPECGAME which models the interactions between the regulator and builder as a Stackelberg game.
- We present a novel algorithm, PARETOPLAY, to simulate SPECGAME, proving it recovers equilibria. Simulations show the cost of strategic behavior can be 70–96% higher compared to collaborative regulation, based on evaluations over six tabular and vision datasets.

## 2 RELATED WORK AND BACKGROUND

**Related Work.** It has been shown that there exist tensions between model accuracy, privacy, and fairness (Tramer & Boneh, 2020; Suriyakumar et al., 2021; Farrand et al., 2020). Attempts to improve the resulting trade-offs have involved adapting the training procedure (Xu et al., 2019; Mozannar et al., 2020; Franco et al., 2021; Tran et al., 2021), a form of hyperparameter search (Avent et al., 2019), or calculating Pareto frontiers (Jagielski et al., 2019; Yaghini et al., 2023). Note that, in contrast to our work, all prior frameworks do not consider the inherent multi-agent nature of the problem: they characterize trade-offs without modeling the regulator (who is enforcing trustworthiness) and the company (who is implementing trustworthiness) as separate entities. While it integrates some of the techniques from prior work, our work innovates by developing the SPECGAME framework to model the interactions between the regulator and company.

For brevity, our paper considers two societal risks: algorithmic bias and leakage of private information. Our framework is more general and extensible to other risks (see Appendix A.1). We now formalize the corresponding definitions of fairness $\Gamma(.)$ and privacy $\mathcal{E}(.)$.

**Fairness.** The choice of fairness measure is largely task-dependent and at the behest of the regulators (Barocas et al., 2018). Hence, our framework abstracts this choice and does not make any assumptions on the applied metric. The fairness evaluation process takes as input a fairness metric $\Gamma_{\text{fair}}(\omega, D) : \mathcal{W} \times \mathcal{X} \mapsto \mathbb{R}^+$ chosen by the regulator, the model $\omega \in \mathcal{W}$, and an adequate evaluation dataset $D_{\text{eval}} \in \mathcal{X}$, $D_{\text{eval}} \sim \mathcal{D}$, where $\mathcal{D}$ is the task's data distribution. The evaluation process then outputs $\widehat{\gamma}_{\omega}$ as an empirical estimate of the model's fairness violation. In Section 4, we instantiate concrete ML algorithms with their stated fairness measures which we discuss in detail in Appendix G.

**Privacy.** In the context of ML, Differential Privacy (DP) (Dwork et al., 2006) adds *controlled noise* to the ML algorithm to protect contributions individuals make to the training set—while still yielding useful models. Our work considers the $(\varepsilon, \delta)$-differential privacy setup. Let $\mathcal{M} : \mathcal{X} \to \mathcal{R}$ be a randomized algorithm. In our case, $\mathcal{M}$ is either the training algorithm or the inference procedure. $\mathcal{M}$ satisfies $(\varepsilon, \delta)$-DP with $\varepsilon \in \mathbb{R}_+$ and $\delta \in [0, 1]$ if for all neighboring datasets $D \sim D'$, *i.e.*, datasets that differ in only one data point, and for all possible subsets $R \subseteq \mathcal{R}$ of the output space it must hold that $\mathbb{P}[\mathcal{M}(D) \in R] \leq e^{\varepsilon} \mathbb{P}[\mathcal{M}(D') \in R] + \delta$.

In our formulations, similar to fairness, we consider a privacy-parameter evaluation function $\mathcal{E}(\omega, D) : \mathcal{W} \times \mathcal{X} \mapsto \mathbb{R}^+$. The evaluation process produces $\hat{\varepsilon}_{\omega}$ as an estimation of the (true) privacy parameter of the model $\varepsilon_{\omega}$. Auditing DP learning is a non-trivial problem due to the worst-case nature of privacy failures (which are intrinsically rare events (Nasr et al., 2021; Chadha et al., 2024)). While our work does not directly contribute to privacy auditing, it benefits from ongoing progress in this area.

**Problem Formulation: Collaborative ML Regulation**

Before we introduce our model for the company to interact with a regulator, possibly strategically, we first need to understand the baseline of voluntary collaboration. In this baseline, which we called COLLABREG in Section 1, the regulator and company form a *committee* that jointly produces a model that balances accuracy and societal risks, i.e., is trustworthy.

Formally, the committee wishes to train a model $\omega \in \mathcal{W}$ on dataset $D \sim \mathcal{D}$ where $\mathcal{D}$ is the data-generating distribution. $\mathcal{W} := \Theta \times \Phi$ is the space of models with $\dim(\Theta)$ parameters and $\dim(\Phi)$ hyper-parameters. Model $\omega$ may have many hyper-parameters, only a subset $\mathcal{S} \subset \Phi$ of which impacts the trustworthy metrics that interests the committee due to their impact on societal risk. We call $\mathcal{S}$ the set of *trustworthy hyper-parameters*. For instance, a convolutional network trained with DP has $\dim(\Phi \setminus S)$ hyper-parameters like filter-size, and $\dim(S) = 2$ trustworthy hyper-parameters, namely the privacy parameters $(\varepsilon, \delta)$ used to train the model (see Section 2). Training such a model is a bi-level optimization problem. The committee first needs to pick trustworthy hyper-parameters $s$; this

is the outer problem. Then, given these trustworthy hyper-parameters $s$, training proceeds as usual to optimize the model parameters and any remaining hyper-parameter; this is the inner problem.

Our focus is on the outer optimization problem. In our running example of a committee training a model that is accurate, fair, and private, the outer problem combines 3 penalties each corresponding to the solution to the inner problem for one of the 3 properties (i.e., accuracy, fairness, or privacy). This loss can be written in vector form as $\boldsymbol{\ell}(\boldsymbol{s}) = [\ell_{comp}(\boldsymbol{s}) \quad \ell_{fair}(\boldsymbol{s}) \quad \ell_{priv}(\boldsymbol{s})]^\top$. Note that computing each of these components involves solving the inner problem, i.e., finding model parameters $\boldsymbol{\theta}^*$ and remaining hyper-parameters $\boldsymbol{\phi}^*$ by training the model. To scalarize $\boldsymbol{\ell}(\boldsymbol{s})$, we introduce a *weighting vector* $\boldsymbol{\lambda} = [1 \quad \lambda_{fair} \quad \lambda_{priv}]^\top$. We obtain the outer problem $\text{minimize}_{\boldsymbol{s} \in \mathcal{S}} \boldsymbol{\lambda}^\top \boldsymbol{\ell}(\boldsymbol{s})$. We note that the weight vector $\boldsymbol{\lambda}$ is a *free parameter* and by varying it we obtain different Pareto optimal solutions to problem (1). *From an algorithmic point of view, all such solutions are valid and none is strictly better than the other.* However, this is not the case from a socioeconomic perspective. We thus need to introduce constraints to the outer problem to indicate socioeconomic requirements of the committee:

$$
\begin{aligned}
\text{minimize}_{\boldsymbol{s} \in \mathcal{S}} \quad & \boldsymbol{\lambda}^\top \boldsymbol{\ell}(\boldsymbol{s}) \\
\text{subject to} \quad & \ell_{comp}(\boldsymbol{s}) \le \alpha, \quad \ell_{fair}(\boldsymbol{s}) \le \gamma, \quad \ell_{priv}(\boldsymbol{s}) \le \varepsilon.
\end{aligned} \tag{1}
$$

The constraint on $\ell_{comp}(\boldsymbol{s}) := \text{err}(\boldsymbol{s})$ represents company's concern regarding the accuracy of their model. A model with error larger than $\alpha$ is not profitable to bring to the market. Since regulation applies only to released models, we implicitly limit our search to those meeting this condition and henceforth omit the explicit constraint. The two other constraints model regulators concerns about societal risk. The fairness regulator measures violations using a fairness metric $\ell_{fair}(\boldsymbol{s}) := \Gamma(\boldsymbol{s})$. A model with violation $\Gamma(\boldsymbol{s}) > \gamma$ is deemed unacceptable. Similarly, the regulator measures privacy cost with $\ell_{priv}(\boldsymbol{s}) := \mathcal{E}(\boldsymbol{s})$ and mandates that $\mathcal{E}(\boldsymbol{s}) \le \varepsilon$.

The optimization problem in Equation (1) encapsulates many prior work in trustworthy ML (*e.g.*, (Zafar et al., 2017)) where a single agent is tasked with optimizing for all objectives. Our main contribution is to consider solutions to the above in a distributed multi-agent setting.

## 3 ML Regulation as a Principal-Agent Problem

We consider ML regulation under the more realistic Principal-Agent setting, where unlike COL-LABREG, there is a separation between the Principal (regulator) and the Agent (company). *This separation means that the regulators and the company can have separate i) goals, ii) knowledge, and iii) actions.* We can formally consider the consequences of this separation in the context of Equation (1). Using its regularized version, we have:

$$
\min_{\boldsymbol{s} \in \mathcal{S}} \mathcal{L}(\boldsymbol{s}) = \min_{\boldsymbol{s} \in \mathcal{S}} \boldsymbol{\lambda}^\top \boldsymbol{\ell}(\boldsymbol{s}) + (\boldsymbol{c} \odot \mathbb{1}_{[\boldsymbol{\ell}(\boldsymbol{s}) \succeq \boldsymbol{b}]})^\top (\boldsymbol{\ell}(\boldsymbol{s}) - \boldsymbol{b}), \tag{2}
$$

where $\boldsymbol{c} = [0 \quad C_{fair} \quad C_{priv}]$ are *penalty scalars* for constraint violations and $\boldsymbol{b} = [1 \quad \gamma \quad \varepsilon]$ the *trustworthy specification* bounds. The Principal-Agent formulation introduces two changes to this objective:

- The possibility of *misaligned incentives* introduced by the separation between the regulators and the company means that they each have their own weighting vectors $\boldsymbol{\lambda}_{reg}$ and $\boldsymbol{\lambda}_{comp}$, respectively. These vectors can be misaligned $\angle(\boldsymbol{\lambda}_{comp}, \boldsymbol{\lambda}_{reg}) \ne 0$.
- Incomplete information can manifest in two ways: *hidden information* and *hidden action*[1].

Hidden information may occur as a result of differences in model architecture and hyper-parameters, but favoring brevity, we focus on *data inequality* as a prototypical example: since two different entities are evaluating Equation (2), the vector objective $\boldsymbol{\ell}(\boldsymbol{s})$ is evaluated on separate datasets $D_{reg}, D_{comp} \sim \mathcal{D}$. Hidden action signals the uncertainty of an entity regarding the other entity's actions: in optimizing Equation (2), the company *may* loosen (or eliminate) the regularization term $(\boldsymbol{c} \odot \mathbb{1}_{[\boldsymbol{\ell}(\boldsymbol{s}) \succeq \boldsymbol{b}]})^\top (\boldsymbol{\ell}(\boldsymbol{s}) - \boldsymbol{b})$. This shows strategic behavior is possible. Thus, the regulator *cannot trust* the company to apply the regularization term. Consequently, after announcing constraints $\boldsymbol{b} = (\gamma, \varepsilon)$ as its trustworthy specification, the regulator chooses to enforce the penalty externally, for instance, as a *monetary fine* (see Appendix A.2 for real-world examples).

---

[1]In agency theory, these are known as *adverse selection* and *moral hazard*, respectively (Alexander, 2006).

Put altogether, the company is forced to solve:

$$\min_{\boldsymbol{s}\in\mathcal{S}} \mathcal{L}_{comp}(\boldsymbol{s}) = \min_{\boldsymbol{s}\in\mathcal{S}} \overbrace{\boldsymbol{\lambda}_{comp}^{\top}\boldsymbol{\ell}(\boldsymbol{s})}^{\text{by the company}} + \overbrace{(\boldsymbol{c} \odot \mathbb{1}_{[\hat{\boldsymbol{\ell}}(\boldsymbol{s})\succeq\boldsymbol{b}]})^{\top}(\hat{\boldsymbol{\ell}}(\boldsymbol{s}) - \boldsymbol{b})}^{\text{by the regulators}}, \quad (3)$$

where the second term is the penalty evaluated by the regulator according to its estimation of the violations of $\hat{\boldsymbol{\ell}}(\boldsymbol{s})$ from the specification bounds $\boldsymbol{b}$. Note that due to the uncertainty in regulators' estimations, even a non-strategic company may get penalized by this hidden information. We will differ formally studying strategic behavior that results from hidden action to Section 3.1. Next, we show that, even in the absence of strategic behavior, hidden information leads to degraded utility.

**Hidden information leads to loss of utility for the company.** If we take the company's perspective, hidden information translates into $\boldsymbol{\ell}(\boldsymbol{s})$ being an incorrect estimation of the regulator's $\hat{\boldsymbol{\ell}}(\boldsymbol{s})$. Recall our running example of a regulator enforcing fairness and privacy. The values of both corresponding penalties can be mis-estimated by the company.

For privacy, estimation uncertainty arises from the fact that the DP parameter $\varepsilon$ is a theoretical upperbound on the true privacy leakage of the model $\omega$: $\varepsilon_- < \varepsilon_\omega < \varepsilon$. The true privacy leakage of the model $\varepsilon_\omega$ depends on training data, and the capabilities of the regulator auditing privacy. Thus, companies estimate a lower bound $\varepsilon_-$ which, together with the theoretical upperbound, provides an estimate on $\varepsilon_\omega$, as well as an uncertainty measure for the privacy leakage of their model $\Delta\varepsilon := \varepsilon - \varepsilon_-$. Research has shown that $\Delta\varepsilon$ can be large relative to $\varepsilon$ (Nasr et al., 2021; Chadha et al., 2024).

Similarly, a large body of work have documented the "instability" of fair classification (Friedler et al., 2018; Huang & Vishnoi, 2020; Cooper et al., 2024) with respect to variations in the training dataset. As a result, black-box audits of fair classifiers can over- and under-estimate the true fairness violations of the model as well. Given the uncertainty regarding fairness and privacy risks of the model, the threat of penalties can lead to over-conservatism by the company. For brevity, we will show this formally for the privacy risk (a similar argument holds for fairness). We can re-write Equation (3) as:

$$\min_{\boldsymbol{s}} \tilde{\mathcal{L}}_{comp}(\boldsymbol{s}) = \min_{\boldsymbol{s}} \text{err}(\boldsymbol{s}) + \lambda_{priv}(\mathcal{E}(\boldsymbol{s}) \pm |\Delta\varepsilon|) + C_{priv}\mathbb{1}_{[\mathcal{E}(\boldsymbol{s})\pm|\Delta\varepsilon|\geq\varepsilon]}(\mathcal{E}(\boldsymbol{s}) \pm |\Delta\varepsilon| - \varepsilon), \quad (4)$$

where we have replaced the estimated privacy leakage of the model $\hat{\ell}_{priv}(\boldsymbol{s}) = \hat{\mathcal{E}}(\boldsymbol{s})$ with $\mathcal{E}(\boldsymbol{s}) \pm |\Delta\varepsilon|$. $\mathcal{E}(\boldsymbol{s})$ is the true privacy leakage of the model and $|\Delta\varepsilon|$ represents the uncertainty of the company both of its own estimation (second term) as well as regulator's estimation (third term). Recall that $\varepsilon$ is the specification mandated by the regulator.

To understand why the company would become over-conservative, consider the following situation. The company is deciding between releasing two models, model 1 is more accurate $\text{err}(\boldsymbol{s}_1) < \text{err}(\boldsymbol{s}_2)$ but has a higher privacy budget than model 2, $\mathcal{E}(\boldsymbol{s}_1) > \mathcal{E}(\boldsymbol{s}_2)$. The decision to release one model or the other is based on the total loss $\tilde{\mathcal{L}}_{comp}(\boldsymbol{s}_1)$ and $\tilde{\mathcal{L}}_{comp}(\boldsymbol{s}_2)$. We seek to find condition under which it is more economically viable to release the less-accurate model 2, *i.e.*, $\tilde{\mathcal{L}}_{comp}(\boldsymbol{s}_2) \leq \tilde{\mathcal{L}}_{comp}(\boldsymbol{s}_1)$.

Seeking worst-case conditions, we consider the case where the company underestimates its own privacy parameter (*i.e.*, second term is $\lambda_{priv}(\mathcal{E}(\boldsymbol{s}) - |\Delta\varepsilon|)$) and regulator underestimates $\hat{\mathcal{E}}(\boldsymbol{s}_2)$ and overestimates $\hat{\mathcal{E}}(\boldsymbol{s}_1)$ such that the third term appears with negative and positive $|\Delta\varepsilon|$, respectively. Furthermore, both values exceed the specification (indicators are 1), therefore: $\text{err}(\boldsymbol{s}_2) - \text{err}(\boldsymbol{s}_1) \leq -(\mathcal{E}(\boldsymbol{s}_2) - \mathcal{E}(\boldsymbol{s}_1))(C_{priv} + \lambda_{priv}) + 2C_{priv}|\Delta\varepsilon|$. We define $\text{UtilityLoss} := \text{err}(\boldsymbol{s}_2) - \text{err}(\boldsymbol{s}_1) \geq 0$ and $\text{PrivacyGain} := -(\mathcal{E}(\boldsymbol{s}_2) - \mathcal{E}(\boldsymbol{s}_1)) \geq 0$ for using model 2 instead of model 1. The condition to incentivize the company to produce a more private but less accurate model is:

$$|\Delta\varepsilon| \geq \frac{1}{2}\left(\frac{\text{UtilityLoss}}{C_{priv}} - \text{PrivacyGain}\left(1 + \frac{\lambda_{priv}}{C_{priv}}\right)\right). \quad (5)$$

In the presence of great uncertainty $|\Delta\varepsilon| \gg 0$, Equation (5) holds regardless of the company's efforts in producing a more private model even at great cost to utility. Similarly, for a large enough $C_{priv}$ chosen by the privacy regulator, the right hand side can be zero (or negative) regardless of utility loss ensuring the inequality holds trivially and forcing the company to always release the less accurate model. In practice, however, as we discussed in see Section 2, the company has an upper bound on the acceptable error of any model they release which means they will not produce a model at all.

**Note.** From Section 2 remember that $\lambda_{priv}$ and $\lambda_{fair}$ form a company's own weighting vector for privacy and fairness losses relative to its error term. Positive $\lambda$s indicate that the company is

Figure 2: **Repeated SPECGAME between Company, and Privacy and Fairness regulators**—regulators-led (top) or company-led (bottom).

| Agent | Cost Function | Strategy |
|---|---|---|
| Fairness Regulator | $\mathrm{cost}_{fair}(\boldsymbol{s}) = \begin{cases} \hat{\Gamma}(\boldsymbol{s}) - \gamma & \hat{\Gamma}(\boldsymbol{s}) \geq \gamma \\ \mathrm{err}(\boldsymbol{s}) & \hat{\Gamma}(\boldsymbol{s}) < \gamma \end{cases}$ | Penalize $L_{fair}(\boldsymbol{s}) =$ $C_{fair}\mathbb{1}_{[\hat{\Gamma}(\boldsymbol{s}) \geq \gamma]}(\hat{\Gamma}(\boldsymbol{s}) - \gamma)$ |
| Privacy Regulator | $\mathrm{cost}_{priv}(\boldsymbol{s}) = \begin{cases} \hat{\mathcal{E}}(\boldsymbol{s}) - \varepsilon & \hat{\mathcal{E}}(\boldsymbol{s}) \geq \varepsilon \\ \mathrm{err}(\boldsymbol{s}) & \hat{\mathcal{E}}(\boldsymbol{s}) < \varepsilon \end{cases}$ | Penalize $L_{priv}(\boldsymbol{s}) =$ $C_{priv}\mathbb{1}_{[\hat{\mathcal{E}}(\boldsymbol{s}) \geq \varepsilon]}(\hat{\mathcal{E}}(\boldsymbol{s}) - \varepsilon)$ |
| Company | $\mathrm{cost}_{comp}(\boldsymbol{s}) = \mathrm{err}(\boldsymbol{s}) + L_{priv}(\boldsymbol{s}) + L_{fair}(\boldsymbol{s})$ | Release model with trustworthy hyper-parameters $\boldsymbol{s}$ |

Table 1: **SPECGAME $\mathcal{G}_b$.** Company releases model with trustworthy hyper-parameters $\boldsymbol{s} \in \mathcal{S}$, regulators issue penalties $L_{fair}, L_{priv} \in \mathbb{R}^+$.

interested in producing trustworthy models even in the absence of regulatory pressure. In the rest of the paper, we will focus on strategic behavior which means that regulators have to assume the worst-case behavior of $\lambda_{priv} = \lambda_{fair} = 0$, *i.e.*, the company is only concerned with its model error. With $\lambda_{fair}, \lambda_{priv} > 0$, our theoretical results remain unaffected because $C_{priv}, C_{fair}$ can be adjusted accordingly to produce the same effect. Appendix F provides guidance to estimate $\lambda$s in practice.

## 3.1 ML REGULATION UNDER STRATEGIC BEHAVIOR

Despite the absence of strategic behavior, incomplete information leads to excessive loss of utility for the company. Conversely, it is possible for the company to take advantage of the uncertainty inherent in risk estimation strategically to produce a more accurate but less trustworthy model. The regulator has to account for this possibility and interact with the company accordingly. To study the outcome of these interactions, we formalize them using a novel game called SPECGAME. We refer the reader to Appendix C for a background on game theory.

We introduce SPECGAME, a game theoretic model of ML regulation that captures the interactions between three agents involved in the life-cycle of an ML model (Tomsett et al., 2018): a *company* who is in charge of producing the model, and two regulators who are in charge of fairness and privacy of the resulting model, respectively. We note that our framework is general and can accommodate other objectives, as long as they are measurable with a loss function. For instance, In Appendix A.1, we show how to use robustness to adversarial examples as an objective. Based on historical precedent and future regulatory plans (see Appendix A.2), we assume regulators are able to give penalties for violations of their respective objectives.

Depending on whether regulators announce trustworthy specifications $\boldsymbol{b}$ (see Section 2) first, or if the company produces a model first with fairness and privacy guarantees of its choosing, we would have a game that is either *regulator-led*, or *company-led* (see Figure 2). In Section 4, we will compare the two setting but since analysis of both are similar, without loss of generality (W.L.O.G), unless otherwise stated, we will assume a regulator-led SPECGAME. This sequential order of interactions lends itself naturally to a *Stackelberg competition* (Fudenberg & Tirole, 1991). In either case, if the company abides by the specification $\boldsymbol{b}$, the game concludes (i.e. the game has a single *stage*). However, if the regulator is not convinced of the company's compliance, the company is penalized and forced to release a new model until the regulator is assured of its compliance[2].

Formally the SPECGAME $\mathcal{G}$ is a repeated Stackelberg game. Its stage game $\mathcal{G}_{\mathrm{stage}} = (\mathcal{A}, \mathcal{S}, \mathcal{C})$ is repeated $T$ times as shown in Figure 2. Each stage is marked with dotted windows. $\mathcal{A} = \{comp, fair, priv\}$ is the set of agents. The strategy space of the stage game is $\mathcal{S} = \{(\boldsymbol{s}_{fair}, \boldsymbol{s}_{priv}, \boldsymbol{s}_{comp})\}$ and $\mathcal{C} = \{(\mathrm{cost}_{fair}, \mathrm{cost}_{priv}, \mathrm{cost}_{comp})\}$ represent agent costs. The complete game is defined as the Cartesian product of the stage game repeated $T$ times: $\mathcal{G} = \mathcal{G}_{\mathrm{stage}}^T$. To analyze $\mathcal{G}$ we are interested in the overall *discounted* cost of agent $i \in \mathcal{A}$ defined as $\overline{cost}_i = \sum_{t=0}^{\infty} c^t cost_i^{(t)}$. $c$ is known as the *discounting factor* and represents the fact that agents care about their cost in the near-term more than in the long run (Shoham & Leyton-Brown, 2009). Table 1 summarizes SPECGAME's agents, their cost functions and strategies which we will elaborate on next:

---

[2]Note this setting also captures other more general settings such as periodic audits, or audits upon release of a new version of the model.

**Regulator cost.** We take the regulator's cost to be of the form $f_{s^*}(\omega) = \begin{cases} \hat{s}_\omega - b & \hat{s}_\omega \geq b \\ \text{err}(\boldsymbol{s}) & \hat{s}_\omega < b \end{cases}$, where

$b \in \{\gamma, \varepsilon\}$ is the regulator's specification for the fairness (or privacy) parameter, $\hat{s}_\omega$ is the regulator's estimation of model's parameter. If the specification is violated ($\hat{s}_\omega > b$), the regulator's loss is the *excessive risk* $\hat{s}_\omega - b$ that the model poses compared to the specification.

**Regulator strategy is to follow the proportionality principle.** Following the *proportionality principle* (Lacey, 2016), which has abundant precedents in regulatory affairs (Allegrezza & Lasagni, 2024), an appropriate strategy for the regulator is to penalize the company proportionally to the excessive risk $\hat{s}_\omega - b$. Thus, the penalty is of the general form $h(\boldsymbol{s}) = C_{reg}\mathbb{1}_{[\hat{s}_\omega \geq b]}(\hat{s}_\omega - b)$, where $C_{reg}, reg \in \{fair, priv\}$ are regulators *penalty scalars*. We saw in Section 3 that large $C_{reg}$ disincentives companies from producing a model at all by posing unnecessarily strict penalties for small violations. The regulator does not seek such an outcome, and in fact prefer to have an accurate model once the specification is met ($\hat{s}_\omega < b$ case). The reason for this is that prior work has shown that inaccurate models have, for instance, worse privacy characteristics (Shokri et al., 2017).

**Company strategy.** The company's cost is a function of its strategy to release a model with trustworthy hyper-parameters $\boldsymbol{s}$:

$$\text{cost}_{comp}(\boldsymbol{s}) = \text{err}(\boldsymbol{s}) + C_{fair}\mathbb{1}_{[\hat{\Gamma}(\boldsymbol{s}) \geq \gamma]}(\hat{\Gamma}(\boldsymbol{s}) - \gamma) + C_{priv}\mathbb{1}_{[\hat{\mathcal{E}}(\boldsymbol{s}) \geq \varepsilon]}(\hat{\mathcal{E}}(\boldsymbol{s}) - \varepsilon). \tag{6}$$

The optimal strategy $\boldsymbol{s}^*$ (aka, the *best response*) of the company is the minimizer of Equation (6). Furthermore, comparing the two equations (3) and (6) reveals that they are indeed the same, hence ***From an optimization perspective, simulating* SPECGAME *is equivalent to distributed (i.e., multi-party) optimization of* COLLABREG.** This new interpretation not only validates our choice of proportional penalties earlier, but also provides a systematic way to estimate penalty scalars $C_{reg}$ using simulated values from a COLLABREG setting. See Appendix B for more details.

### Solving SPECGAME

A single-stage SPECGAME is a Stackelberg competition analyzing which involves solving a bi-level min-max optimization problem where the follower's feasible strategies are limited by the leader's chosen strategy. The solutions to this problem produce *Stackelberg equilibria*. In the repeated setting, visualized as a tree (akin to a decision-tree) in Figure 2, the appropriate equilibrium concept is a *subgame-perfect equilibrium (SPE)* which requires that the solution produces an equilibrium at every sub-game associated with a sub-tree. Both Stackelberg and subgame-perfect equilibria are extensions of *Nash equilibria* (see Appendix C) to extensive-form games.

However, although Nash equilibria are optimal w.r.t. single-agent deviations, they are often not Pareto efficient. For instance, seeking NEs can provide 'solutions' where both the company and a regulator's losses can be improved simultaneously which is not a desirable outcome for ML regulation. Furthermore, the SPECGAME described in Section 3.1 cannot be simulated directly due to challenges in forming the agents' loss functions, notably, because privacy violations of a trained model is difficult to estimate without access to its training procedure (Gilbert & McMillan, 2018). In Section 3.2 we introduce PARETOPLAY to address these problems by taking advantage of the fact that agents estimate losses using different datasets sampled from the same distribution (see Appendix A.3).

### 3.2 PARETOPLAY: BEST-RESPONSE PLAY ON THE PARETO FRONTIER

In PARETOPLAY, each agent has access to their own Pareto frontier. Companies can easily calculate Pareto frontier from their training checkpoints, but regulators must obtain theirs through a third party (*e.g.*, public data) or the company. In the latter case, cryptographic methods like homomorphic encryption can ensure data privacy during this process. The specifics of regulatory data access are beyond this work's scope but it is a crucial issue that is relevant beyond ML. For instance, in environmental regulation, companies often voluntarily (Bier & Lin, 2013) provide data to reduce detrimental effects of regulator's uncertainty in risk estimation (see Section 3).

The game starts by distributing an initial Pareto frontier between all agents. The Pareto frontier is formed by training multiple instances of the chosen ML models in $R = \{(\text{err}(\boldsymbol{s}), \Gamma(\boldsymbol{s}), \mathcal{E}(\boldsymbol{s})) \mid \boldsymbol{s} \in \mathcal{S}\}$ before the game using different guarantee levels $\boldsymbol{s} := (\gamma, \varepsilon)$ and then calculating the Pareto frontier $PF_i : \mathcal{S} \mapsto [0, 1] \times [0, 1] \times \mathbb{R}^+$ a map from trustworthy parameters to a tuple of achieved error, fairness and privacy losses.

Assuming regulators lead, they select a point on the Pareto frontier. That is, their initial strategy is to play the specification $s^{(0)} = b = (\gamma, \varepsilon)$ which decides the trade-off between fairness and privacy that the regulators seek. In the next round, the company takes a gradient step to improve its error (Line 8). If the updated parameters violate the specification, they penalize the company by taking a gradient step to reduce trustworthy violations (Line 6). Since these updates take the $s^{(t)}$ in opposing directions, PARETOPLAY is a variant of Gradient Ascent-Descent (GDA) algorithm commonly used to solve such bi-level optimization problems (Goktas & Greenwald, 2022).

---

**Algorithm 1 PARETOPLAY**: Regulator-led

**Input:** Trustworthy specification $b$, Initial Pareto frontier inputs $R_i^{(0)}$, $i \in N = \{comp, fair, priv\}$, total number of game rounds $T$, Regulator penalty scalars $C_{fair}, C_{priv}$, step size $\eta$

1: **for** $t \in \{0, 1, \dots, T\}$ **do**
2:     $P_i \leftarrow \mathrm{PF}(R_i^{(t)} \cup \{\tilde{R}\})$    ▷ Agents estimate Pareto frontiers
3:     **if** t = 0 **then**             ▷ First round of the game
4:        $s^{(0)} \leftarrow b$
5:     **else if** $t \mod 2 = 0$ **then**      ▷ Regulators move
6: 
$$s^{(t+1)} \leftarrow s^{(t)} - \eta \left( e_{fair} \odot \nabla_s L_{fair}(s^{(t)}, C_{fair}; P_{fair}) \right.$$
$$\left. + e_{priv} \odot \nabla_s L_{priv}(s^{(t)}, C_{priv}; P_{priv}) \right)$$
7:     **else**                ▷ Company move
8:        $s^{(t+1)} \leftarrow s^{(t)} - \eta \nabla_s \mathrm{err}(s^{(t)}; P_{comp})$
9:     $\tilde{R} \leftarrow \mathrm{CALIBRATE}(s^{(t+1)})$
10:     $\eta \leftarrow c \cdot \eta$       ▷ Agent discounts its payoff by $c$
11: **Output** $s^{(T)}$

---

In PARETOPLAY, we estimate all agent losses on their Pareto frontier $P_i$. Our estimation involves a linear interpolation on $P_i$. Interpolation may lead to estimation errors, as the estimated next parameters $s^{(t+1)}$ may, in fact, not be on the Pareto frontier. We avoid this by including a *calibration* step at the end of each round. $\mathrm{CALIBRATE}(:)\mathcal{S} \mapsto [0, 1] \times [0, 1] \times \mathbb{R}^+$ is a function that takes input trustworthy parameters $s^{(t+1)} \in \mathcal{S}$ where $\mathcal{S}$ is the space of trustworthy hyper-parameters, trains a model using $s^{(t+1)}$ on the agent's dataset, and measures its achieved error $\mathrm{err}(.)$ in $[0, 1]$, fairness violations $\Gamma(.)$ in $[0, 1]$ and privacy parameter $\mathcal{E}(.)$ in $\mathbb{R}^+$ and returns the tuple $\tilde{R} = (\mathrm{err}(s^{(t+1)}), \Gamma(s^{(t+1)}), \mathcal{E}(s^{(t+1)}))$. The next player will recalculate a potentially improved Pareto frontier with the new result $\tilde{R}$ (line 2). Next, we introduce the equilibrium concept that simulating SPECGAME using PARETOPLAY induces.

**Game Theoretic Analysis of SPECGAME under PARETOPLAY.** Playing on the Pareto frontiers has important implications for the equilibrium search: the Pareto frontier gives a signal to every player what to play (similar to how a stop-light allows drivers to coordinate when to pass an intersection). This is known as a *correlation device*. If playing according to the signal is a best response for every player, we recover a correlated equilibrium (see Appendix C). Since SPECGAME is potentially repeated we require an extension to this concept. An *extensive-form correlation device* sends separately and confidentially message $M_i$ to each players $i \in N = \{comp, fair, priv\}$ at the beginning of each stage (*i.e.*, each player samples their own dataset and train their own models).

Formally, the extensive-form correlation device $Q$ consists of messages in the form of Pareto frontiers $M_i = PF_i$ over the objectives, and a probability distribution $\mu$ on the Cartesian product of these message sets $M = \underset{i \in N}{\times} M_i = \underset{i \in N}{\times} PF_i$ where the randomization is over the datasets $D_i = (X_i, Y_i) \sim \mathcal{D}$ used to hyper-parameter tune each model. $\mathcal{D}$ is the data-generating distribution of input features $X_i \in \mathcal{X}$ and labels $Y_i \in \mathcal{Y}$. Using $Q$ in Appendix D we prove:

**Theorem 1.** PARETOPLAY *recovers the Subgame Perfect Correlated Equilibria (SPCEs) of* SPECGAME.

While SPECGAME models ML Regulation as a PAP, it also subsumes COLLABREG as a special case. Indeed, if the company released a model that satisfies the specification $b$ the game converges in one step and no penalty is issued. However, if the game goes on for several stages, both players sustain accumulating losses in the form of penalties (for the company) and untrustworthy models released to the public (for the regulator). Thus, compared to COLLABREG, SPECGAME is inherently inefficient.

*Price of Anarchy (PoA)* (Koutsoupias & Papadimitriou, 1999) is the canonical measure for quantifying the inefficiency caused by strategic self-interested behavior. Given a game (SPECGAME), a notion of equilibrium (SPCE) and a non-negative *group-cost* function (*e.g.*, sum of all agents' costs), *the PoA of the game is defined as the ratio between the largest group-cost of an equilibrium and the group-cost of an optimal outcome*—which in our case is a COLLABREG outcome. We differ a formal definition of our PoA to Appendix E and leave upper-bounding PoA for SPECGAME to future work. In the

next section, we empirically estimate PoA through repeated simulations of the game, offering a lower bound on PoA that remains a useful measure of inefficiency.

## 4 EMPIRICAL EVALUATIONS

**Summary.** We empirically verify our claims in Section 2 regarding excessive loss of utility due to imperfect information (up to 8% in Figure 4). We report the empirical price of anarchy in Table 2 suggesting strategic behavior in SPECGAME results in 70–96% higher group cost compared to COLLABREG. Next, we evaluate the usefulness of SPECGAME simulated via PARETOPLAY as a virtual sandbox for ML regulators. Notably, we show it benefits regulators to take initiative in specifying regulations (reducing privacy parameter $\varepsilon$ by up to 6 in Table 3). Given the universal impact of incomplete information, we verify that regulators can enforce compliance with their specification even when they estimate their Pareto frontier on different datasets (Figure 5). We share additional results in Appendix H.

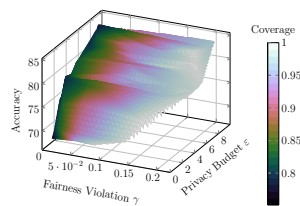

Figure 3: **Pareto frontier example for UTKFace using Fair-PATE.** Akin to Figure 1.

**Algorithm.** We instantiate PARETOPLAY with Fair-PATE (Yaghini et al., 2023), which trains fair and private classification models. It uses demographic parity as its fairness notion requiring equalized prediction rates between different subgroups. As is customary in DP training, we set $\delta = 10^{-6}$ according to the dataset size. We define $s_{\text{FairPATE}} = (\gamma, \varepsilon)$, where $\gamma$ is the maximum tolerable demographic disparity between any two subgroups, and $\varepsilon$ is the differential privacy budget. Furthermore, FairPATE produces classifiers with a *reject option* (Cortes et al., 2016) which means that the classifier can reject answering queries (instead of producing inaccurate, or in this case, unfair) decisions. We measure "coverage" as another utility metric in addition to accuracy: coverage is the percentage of queries answered by the model at inference. Higher coverage is better as rejection can also come at a cost (of invoking another model or deferring prediction to a human). Figure 3 depicts FairPATE's Pareto frontier on UTK-Face. We run each experiment with 5 different specification $b$ and aggregate the results. All results are plotted with 95% confidence intervals (CI). We defer details to Appendix F.1.

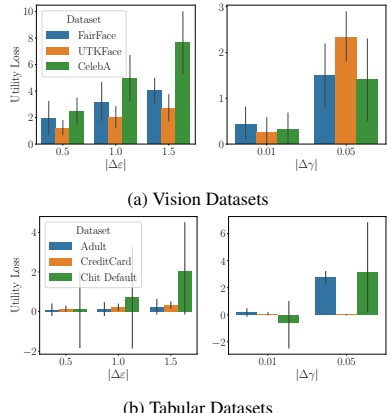

(a) Vision Datasets

(b) Tabular Datasets

Figure 4: **Uncertainty in privacy estimation causes up to a 8% reduction in utility for vision data, and 4% for tabular data.** Uncertainty in estimation of fairness has a negligible impact.

**Datasets.** We adopt the experimental setup of Yaghini et al. (2023) for FairPATE. We perform gender classification on UTKFace (Zhang et al., 2017) and Fairface (Karkkainen & Joo, 2021) datasets where "race" is the sensitive attribute. On CelebA (Liu et al., 2015) the classification task is "whether the person is smiling" and "gender" is the sensitive attribute used for evaluating the fairness constraint. We also report results on 3 tabular datasets where "gender" is the sensitive attribute. In Taiwan Credit Card (Yeh, 2009) and Chit Defaults (Rao, 2018) we predict "whether the person will default on their payment in the next month." In Adult (Becker & Kohavi, 1996), we predict "whether the individual will make more than $50K."

**SPECGAME and PARETOPLAY Settings.** All games are regulator-led unless otherwise specified. As noted in Section 3, we set $\lambda_{priv} = \lambda_{fair} = 0$. We systematically estimate $C_{priv}$ and $C_{reg}$ for each dataset using the procedure detailed in Appendix B.2 and report values in Appendix I. We use the discounting factor $c = 0.67$.

**Excessive Utility Loss and Price of Anarchy.** We estimate company's utility loss in terms of accuracy and coverage due to hidden information using pre-computed Pareto frontiers on tabular and vision data (Figure 4). To avoid penalties, the company needs to take uncertainty into account and thus produces models that follow stricter constraints. Note that in this experiment we are not considering

| Dataset | Price of Anarchy |
|---|---|
| UTKFace | $1.96 \pm 0.10$ |
| CelebA | $1.80 \pm 0.40$ |
| FairFace | $1.71 \pm 0.34$ |
| Adult | $1.75 \pm 0.02$ |
| CreditCard | $1.70 \pm 0.06$ |
| Chit Defaults | $1.83 \pm 0.08$ |

Table 2: **PoA in SPECGAME.** Strategic behavior causes group cost (sum loss of all players) 70–96% higher w.r.t. COLLABREG.

| Metric (company-led− regulator-led) | UTKFace | CelebA | FairFace | Adult | CreditCard | Chit Defaults |
|---|---|---|---|---|---|---|
| Privacy Budget $\varepsilon$ ($\downarrow$) | $3.97 \pm 2.40$ | $3.47 \pm 1.40$ | $5.95 \pm 1.95$ | $0.54 \pm 0.21$ | $-0.06 \pm 0.25$ | $-0.06 \pm 0.39$ |
| Disparity $\gamma$ ($\downarrow$) | $0.01 \pm 0.03$ | $0.0 \pm 0.04$ | $0.05 \pm 0.03$ | $0.01 \pm 0.01$ | $0.0006 \pm 0.0007$ | $0.01 \pm 0.02$ |
| Accuracy ($\uparrow$) | $4.37 \pm 3.39$ | $2.01 \pm 1.48$ | $5.77 \pm 8.64$ | $0.05 \pm 0.09$ | $0.09 \pm 0.08$ | $0.01 \pm 0.15$ |
| Coverage ($\uparrow$) | $4.10 \pm 6.03$ | $-3.01 \pm 3.79$ | $4.70 \pm 7.06$ | $0.73 \pm 1.29$ | $0.04 \pm 0.03$ | $0.72 \pm 1.27$ |

Table 3: **First-mover has an advantage in SPECGAME.** We compare a company-led game to a regulator-led one and show the differences in objective values. The 95% CIs are taken over 5 different initial specifications.

strategic behavior and the loss of utility is purely due to estimation uncertainty (see Section 3). We observe that uncertainty in privacy estimation has a large impact on accuracy while the effect is much more subdued for fairness. Uncertainty of $\Delta\varepsilon = 1.5$ can cause up to 8% drop in utility. We also measure price of anarchy when company instead takes advantage of the uncertainty to produce models that violate constraints but have higher utility (Table 2). We calculate group cost using formulation from Section 3.1. We report averaged $PoA_{\boldsymbol{b}}$ over 5 different initial specifications $\boldsymbol{b}$ as well as 95% confidence intervals. On all six datasets, strategic behaviour leads to group costs that are 70–96% higher compared to that in collaborative regulation.

**SPECGAME leader has a first-mover advantage.** Recall that in each game, the first-mover chooses the point on the Pareto surface that minimizes their loss. All other parameters in both games, including regulators' fairness and privacy constraints, remain the same throughout the game run. In Table 3, we show the difference in achieved objective values changing from a regulator-led game to an company-led one. On vision datasets, when the company leads, it produces models that are on-average 5 percentage points more accurate (a) and answer 5 percentage points more queries (b) compared to when the regulator leads; however, this comes at the cost of a minor 0.02 increase in disparities (c) and a large privacy budget increase of 4 (d). Therefore, regulators should take initiative in making their specifications. We note that we observe a much weaker first-mover advantage on tabular data.

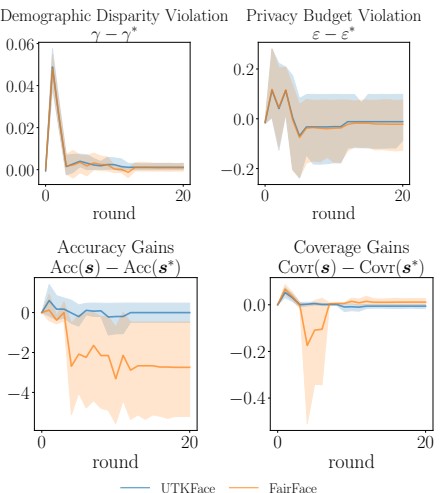

Figure 5: **Agents can have separate datasets in PARETOPLAY.** We simulate a regulator-led game where regulators have access to FairFace and the company has access to UTKFace. The resulting company's model has on average 2% higher accuracy compared to the regulator's. Despite these differences, SPECGAME converges and follows a similar trajectory for both agents in terms of privacy and fairness violations.

**Information equality is not necessary for PARE-TOPLAY.** In Figure 5, regulators have access to FairFace, whereas the company has access to UTK-Face. The agents then use their respective dataset to form their loss functions. Each trains and calibrates their own model on their own datasets. The company's model has on average 2% higher accuracy compared to the regulator's. However, SPECGAME converges and follows a very similar trajectory for both agents in terms of privacy and fairness violations—ensuring that regulator specifications are generally satisfied. We observe similar trends for tabular data (see Figure 9 in Appendix H).

## 5 DISCUSSION & FUTURE WORK

Our approach recognizes the diverse nature of agents involved in deploying and auditing ML models. This allows us to make suggestions for guarantee levels that are more likely to be realizable in practice; given that the gains and benefits of different parties have been taken into account. That said, we made assumptions regarding the economic model under which we operate. While these assumptions follow established principles in economics and in ML, both are contested in their respective literature. We discuss other limitations of our approach in further details in Appendix J.

Furthermore, we centered our consideration around calculating fines proportional to the privacy and fairness violations of chosen guarantee levels $(\gamma, \varepsilon)$; as well as ensuring they are effective in changing company behavior. SPECGAME instantiates the idea of a virtual sandbox, which we mentioned when opening our manuscript. Deploying this idea in the real world is of course a natural next step. Finally, the converse problem is also important: assuming a bound $C$ on the penalty, what are the maximal $\gamma, \varepsilon$ guarantees that we can expect to be able to enforce?

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

## A    MODELING DECISIONS AND JUSTIFICATIONS

### A.1    SOCIETAL RISKS BEYOND FAIRNESS AND PRIVACY

SPECGAME is extensible to include other trustworthy objectives. For example, consider *robustness to adversarial examples as an objective*. The regulator can produced perturbed examples that successfully fool a model to change its prediction. Given the transferability of adversarial examples, the regulator can then audit the company's model and produce an attack success rate $a_\epsilon(\omega) \in [0, 1]$ given a maximum perturbation of size $\epsilon$. There is a large body of work that produces certifications for robustness to adversarial examples (see Cohen et al. (2019)). These are typically of the form $\|a_\epsilon(\omega)\| \leq c$. In the new SPECGAME, the regulator can produce a specification $\|a_\epsilon(\omega)\| \leq c^*$ for a given $\epsilon = \epsilon_{reg}$. The regulator then audits the company model $\omega_{comp}$, and estimates $\hat{a}_{\epsilon_{reg}}(\omega_{comp})$. The penalty assigned by the regulator is of the form $C \cdot \mathbf{1}[\hat{a}_{\epsilon_{reg}}(\omega_{comp}) > c^*](\hat{a}_{\epsilon_{reg}}(\omega_{comp}) - c^*)$.

## A.2 Regulatory Penalties in the Real-World

In the past, regulators often issue penalties for fairness and privacy violations. Concretely, for data privacy GDPR Enforcement Tracker) tracks the violations and fines issued for GDPR non-compliance. Similarly, the Federal Housing Administration (FHA) has frequently issued penalties for violations of Fair Housing Act — a key application scenario in algorithmic fairness research:

> Respondents who have violated the Fair Housing Act in the previous 5 years can be fined a maximum of \$54,157.00. Respondents who have violated the Act two or more times in the previous 7 years can be fined a maximum of \$108,315.00.

Furthermore, Article 99: Penalties of the newly established EU AI Act clearly establishes penalties for non-compliance with "(e) obligations of deployers pursuant to Article 26;" for the "deployers of high-risk AI systems." Therefore, *we base our assumption that "regulators are able to give penalties" on both historical precedent and future regulatory plans.*

## A.3 On the Similarity of the Pareto frontiers

We show that it is not necessary to assume that the Pareto frontiers of the company and the regulators are the same. Rather, it is enough to assume that the datasets they are calculated on are from the same data-generating distribution. Concretely, we show that, the problem of finding the Pareto frontier for each agent can be written as a multi-objective optimization problem, the solution to which reduces to empirical risk minimization in ML. We conclude that the assumption of the shared Pareto frontier between agents is akin to the standard assumption of IID-ness (independent and identically-distributed data) in ML.

**Deriving the Pareto frontier via scalarization.** There exist standard techniques to recover the Pareto frontier of a multi-objective optimization problem—which always exists for any feasible problem. *Scalarization* (Boyd & Vandenberghe, 2004, Section 4.7.4) is such a technique that, provided each objective is convex, can recover all of the Pareto frontier; and if not, at least a part of it. For our problem, the objective loss of the scalarized problem is $\min_{\boldsymbol{s}} \alpha_1 \ell_{comp}(\boldsymbol{s}) + \alpha_2 \ell_{fair}(\boldsymbol{s}) + \alpha_3 \ell_{priv}(\boldsymbol{s})$, where $\alpha_1, \alpha_2, \alpha_3 \geq 0$ are free parameters, different choices for which will give us various points on the Pareto frontier. Implicit in the scalarized objective loss are two assumptions: a) the dataset used to optimize the loss, and b) dependency on model weights $\omega$. Making these assumptions explicit allows us to write the Pareto frontier $PF_i$ calculated by agent $i$:

$$PF_i = \{\arg\min_{\boldsymbol{s}} \ \min_{\omega} \alpha_1 \ell_{comp}(\boldsymbol{s}, \omega; D_i) + \alpha_2 \ell_{fair}(\boldsymbol{s}, \omega; D_i)$$

$$+ \alpha_3 \ell_{priv}(\boldsymbol{s}, \omega; D_i) \mid \alpha_1, \alpha_2, \alpha_3 \in \mathbb{R}^+\}, \tag{7}$$

where $PF_i$ is calculated over dataset $D_i$ by agent $i$. Seen through an ML lens, Equation (7) closely resembles an empirical risk minimization (ERM) problem. We optimize model parameters $\omega$ in the inner sub-problem and tune the hyper-parameters $\boldsymbol{s}$ in the outer one.

Coming back to question of whether Pareto frontiers are similar for different agents, we argue that since the problem of finding in the Pareto frontier reduces to an ERM problem, despite $D_i$ not being the same, we expect that the Pareto frontiers would be similar provided that $D_i \sim \mathcal{D}$ where $\mathcal{D}$ is the data-generating distribution, and that each $D_i$ have enough samples. In other words, *the true correlation device in* PARETOPLAY *is not so much the Pareto frontier, but the real-world phenomenon whose data is sampled by each agent.*

We conclude this section by noting that prior works supports our assumption as well. Yaghini et al. (2023, Section 5.1.4) empirically showed that the Pareto frontiers calculated on separate datasets but for the same task are quite similar. In Section 4, we empirically evaluate the shared Pareto frontier assumption. We simulate a SPECGAME using PARETOPLAY where regulators and companies use different datasets but for modeling the same task (gender estimation). PARETOPLAY converges because all agents are modeling the same data-generating phenomenon (gendered humans).

# B Incentive Design: Choosing Penalty Scalars

Choosing appropriate penalty scalars $C_{reg}$ is crucial for effective regulation. Small values can make the regulation ineffective by turning the monetary penalty into a cost of business and having no

effect on the trustworthiness of the models the company releases, while overly large values of $C_{reg}$ can disincentives releasing a model at all as we saw in Section 3. Since each choice of $C_{reg}$ produces a game with a particular equilibrium, our focus here is to help the regulator design $C_{reg}$ to induce a desirable equilibrium. In the algorithmic game theory literature, this is known as incentive (mechanism) design.

Intuitively, penalty scalars $C_{reg}$ are chosen to be large enough to offset economical gains from producing an untrustworthy model. This is easy using a similar calculation that led to Equation (5). Here we seek to find under what conditions the company would prefer to release the more accurate $\text{err}(s_1) < \text{err}(s_2)$ model 1 instead of the more private model 2 $\mathcal{E}(s_2) < \mathcal{E}(s_1)$. As discussed in Section 3, the company makes that decision based on its total loss $\tilde{\mathcal{L}}_{comp}(s_1) \leq \tilde{\mathcal{L}}_{comp}(s_2)$. Defining UtilityGain := $-(\text{err}(s_1) - \text{err}(s_2)) \geq 0$ and PrivacyLoss := $(\mathcal{E}(s_1) - \mathcal{E}(s_2)) \geq 0$ from using model 1 instead of model 2, and setting $\lambda_{priv} = 0$ (for a worst-case analysis) we have:

$$C_{priv} \geq \frac{\text{UtilityGain}}{\text{PrivacyLoss} - 2|\Delta \varepsilon|} \tag{8}$$

Note how uncertainty $|\Delta \varepsilon| > 0$ bloats the penalty scalar; which leads to over-conservative regulation (see Section 1).

In the rest of this section, we present more systematic ways to estimate effective penalty scalars. In Appendix B.1 we find optimal values for $C_{reg}$ using Lagrangian multipliers. In Appendix B.2 we use the connection we established between simulating SPECGAME and solving COLLABREG to estimate appropriate values for $C_{reg}$ using a similar method to Appendix B.1.

### B.1 OPTIMAL PENALTY SCALARS UNDER COLLABREG ARE LAGRANGIAN MULTIPLIERS

The joint committee of regulator-company can solve Equation (1) using Lagrangian optimization because we are in the collaborative setting COLLABREG. Defining $b = [\gamma \quad \varepsilon]$ and Lagrangian multipliers $\nu = [\nu_{fair} \quad \nu_{priv}]$ and adopting $\succeq$ for comparing vectors element-wise, the Lagrangian is $\mathcal{L}(s, \nu) = \lambda^\top \ell(s) + (\nu \odot \mathbb{1}_{[\ell(s) \succeq b]})^\top (\ell(s) - b)$. The KKT conditions Karush (1939); Kuhn & Tucker (1951) for optimal primal $s^*$ and dual $\nu^*$ to the problem are: i) primal feasibility $\ell(s^*) \preceq b$ , ii) dual feasibility $\nu^* \succeq 0$ , and iii) first-order optimality condition $\nabla_s \mathcal{L}(s^*, \nu^*) = 0$. Note that by including the indicator in the Lagrangian, we have also ensured complementarity Boyd & Vandenberghe (2004). In practice, we can use trust region methods to calculate primal dual optimal $s^*, \nu^*$ Conn et al. (2000).

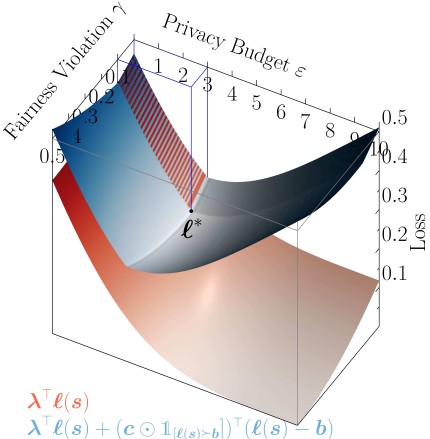

$\lambda^\top \ell(s)$
$\lambda^\top \ell(s) + (c \odot \mathbb{1}_{[\ell(s) \succeq b]})^\top (\ell(s) - b)$

Figure 6: **COLLABREG.** Unconstrained surface in red and regularized surface in blue. The shared surface is the feasible set where trustworthy constraints (blue box) are met. The optimum $\ell^*$ occurs at the boundary of overlap.

To illustrate what such a Lagrangian solution would look like, let us consider a particular scenario where we drop the constraint $\ell_{comp}(s) \leq \alpha$ on model error but requiring that $\lambda = [1 \quad 0 \quad 0]^\top$. This comes without loss of generality because the corresponding penalties are still being enforced through the constraints in Equation (1). Figure 6 depicts the Pareto surface $x = \ell_{fair}(s)$, $y = \ell_{priv}(s)$, and $z = \ell_{comp}(s)$. The red surface shows the unconstrained problem $\text{minimize}_{s \in \mathcal{S}} \ \lambda^\top \ell(s)$.

The constraints from Equation (1) are visualized using the blue bounding box $\ell(s) \preceq b, b = [0.1, 3]$. The optimum occurs at the boundary of the constraints $\ell^* = b$. Intuitively, the joint committee picks the points that maximally uses the tolerated violation of fairness and privacy to obtain the highest possible accuracy. The blue surface shows the equivalent regularized problem $\min_s \mathcal{L}(s, \nu^*)$ with the optimal dual variables $\nu^*$. The overlapping region highlights the feasibility set of the primal problem. This is the region where the constraints are met (indicator is 1). In the rest of the region, the constraints are active, explaining the gap between the two surfaces.

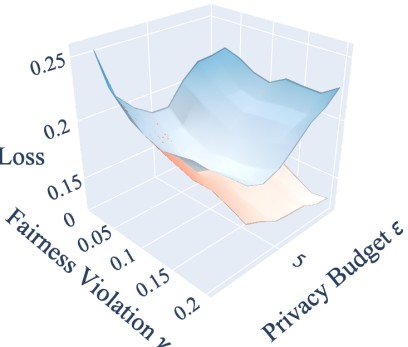

Figure 7: **Unconstrained and constrained surface on UTKFace**. The red surface is unconstrained and the blue is constrained. The corresponding constraints used are $b = (0.1, 4)$.

We define *penalty scalars* $c = [C_{fair} \quad C_{priv}]$ as the optimal Lagrangian multipliers: $c = \nu^*$. We were able to derive them in COLLABREG because $\ell(s)$ is known to every committee member with certainty—conditions that do not hold in SPECGAME. We study the consequences of this next.

### B.2 ESTIMATING PENALTY SCALARS $C_{fair}, C_{priv}$ DESPITE INCOMPLETE INFORMATION

Consider the Principal-Agent setting where the company and regulators are separate agents. This is the setting SPECGAME adopts. In this setting, agents have access to incomplete information. From the regulator's perspective, they do not know the true value of $\ell_{comp}(s)$, which makes selecting appropriate penalty scalars more difficult. Nonetheless, we are still able calculate $C_{fair}$ and $C_{priv}$ according to method detailed in Appendix B.1 with estimations.

---

**Algorithm 2** Estimating $C_{fair}, C_{priv}$

---

**Require:** List of N input model specifications: $s = [(\gamma, \varepsilon)]^N$, list of corresponding N model output losses on the Pareto Frontier: $\ell = [(\ell_{fair}, \ell_{priv}, \ell_{comp})]^N$, desired model specifications $s = (\gamma, \varepsilon)$
    **Step 1:** Train a polynomial regression model to predict $\ell_{comp}$ using $\ell_{fair}$ and $\ell_{priv}$.
    $f(\ell_{fair}, \ell_{priv}) \leftarrow \text{RegressionModel}(\ell)$

    $C_{fair} \leftarrow [\,]$                                         ▷ Initialize lists to store the C values
    $C_{fair} \leftarrow [\,]$
    **Step 2:** Calculate Lagrangian multipliers.
    $Grid(\ell_{fair}, \ell_{priv}) \leftarrow \text{Meshgrid}(\ell)$     ▷ Create a grid of points in the range of input $\ell_{fair}, \ell_{priv}$

    **for** $(\ell_{fair}, \ell_{priv}) \in Grid$ **do**                   ▷ Iterate over all points in the grid.
        $C_{fair}^i, C_{priv}^i \leftarrow \text{TrustConstrained}(f, (\ell_{fair}, \ell_{priv})^i, bounds = [(0, \gamma), (0, \varepsilon)])$
        $C_{fair} \leftarrow C_{fair} + [C_{fair}^i]$
        $C_{priv} \leftarrow C_{priv} + [C_{priv}^i]$
    $C_{fair}^{final} \leftarrow \text{Average}(C_{fair})$
    $C_{priv}^{final} \leftarrow \text{Average}(C_{priv})$
    **Output** $C_{fair}^{final}, C_{priv}^{final}$

---

As mentioned, unlike in collaborative regulation setting, regulators do not have access to exact $\ell_{comp}(s)$ in SPECGAME. However, they can approximate it by training models, obtaining a Pareto frontier, and then estimating $\ell_{comp}(s)$ on the Pareto frontier.

We show the implementation in Algorithm 2. We first train a second degree polynomial regression model using points on pre-computed Pareto frontier to approximate $\ell_{comp}(s)$. $C_{fair}$ and $C_{priv}$ are dependent on $b$, so we calculate them separately for each set of regulators' constraints $b$. We create a grid of points within the fairness and privacy range of the Pareto frontier surface. Then starting

at each point, we use the trust-region constrained algorithm to calculate the Lagrangian multipliers required to enforce the constraint $b$. We set $C_{fair}$ and $C_{priv}$ to the average of calculated Lagrangian multipliers.

Figure 7 shows an example of constrained surface on UTKFace with $b = (0.1, 4)$. The red surface is unconstrained and the blue is constrained. It is transformed by applying the penalties with calculated optimal $C_{fair}$ and $C_{priv}$. We see that $\gamma = 0.1$ and $\varepsilon = 4$ is the lowest point on the blue surface.

## C BACKGROUND ON GAME THEORY

We introduce the following background on game theory from Roth (2017):

**Definition 1** (Mixed Nash Equilibrium). *A mixed strategy Nash equilibrium is a tuple $p = (p_1, \ldots, p_n) \in \Delta A_1 \times \ldots \times \Delta A_n$ such that for all $i$, and for all $a_i \in A_i$ :*

$$u_i(p_1, p_{-i}) \geq u_i(a_i, p_{-i}),$$

*where $p_i \in \Delta A_i$ is a probability distribution over actions $a_i \in A_i$: i.e., a set of numbers $p_i(a_i)$ such that, 1) $p_i(a_i) \geq 0$ for all $a_i \in A_i$, 2) $\sum_{a_i \in A_i} p_i(a_i) = 1$. For $p = (p_1, \ldots, p_n) \in \Delta A_1 \times \ldots \times \Delta A_n$, we write: $u_i(p) = E_{a_i \sim p_i}[u_i(a)]$.*

Searching for Nash equilibria (NEs) is NP-hard (Daskalakis et al., 2006), which is why a super-set of them, known as *Correlated equilibria* have seen increasing attention due to ease with which they can be found (for instance, using polynomial weights algorithm) (Arora et al., 2012; Nisan et al., 2007).

**Definition 2.** *Correlated Equilibrium A correlated equilibrium is a distribution $\mathcal{D}$ over action profiles $A$ such that for every player $i$, and every action $a_i^*$ :*

$$E_{a \sim \mathcal{D}}[u_i(a)] \geq E_{a \sim \mathcal{D}}[u_i(a_i^*, a_{-i}) \mid a_i]$$

Intuitively, A correlated equilibrium is a distribution over action profiles $a$ such that after a profile $a$ is drawn, playing $a_i$ is a best response for player $i$ conditioned on seeing $a_i$, given that everyone else will play according to $a$.

**Definition 3.** *The best-response to a set of actions $a_{-i} \in A_{-i}$ for a player $i$ is any action $a_i \in A_i$ that maximizes $u_i(a_i, a_{-i})$ :*

$$a_i \in \arg\max_{a \in A_i} u_i(a, a_{-i})$$

In multi-objective optimization, and games in particular, we are interested in the *Pareto efficiency*. A tuple of objective values are Pareto efficient if we cannot improve one of the values without making another worse off. More formally, given objectives parameterized by ML models, we have:

**Definition 4** (Pareto Efficiency). *A model $\omega \in \mathcal{W}$, where $\mathcal{W}$ is the space of all models, is Pareto-efficient if there exists no $\omega' \in \mathcal{W}$ such that **(a)** $\forall i \in \mathcal{L}$ we have $\ell_i(\omega') \leq \ell_i(\omega)$ where $\mathcal{L}$ is the set of losses and $\ell_i \in \mathcal{L}$ is the objective $i$'s loss; and that **(b)** for at least one loss $j \in \mathcal{L}$ the inequality is strict $\ell_j(\omega') < \ell_j(\omega)$.*

### C.1 STACKELBERG COMPETITIONS

Stackelberg competitions model sequential interaction among strategic agents with distinct objectives Fudenberg & Tirole (1991). They involve a leader and a follower. The leader is interested in identifying the best action (BR) assuming rational behavior of the follower. The combination of the leader's action and the follower's rational best reaction leads to a strong Stackelberg equilibrium (SSE) Birmpas et al. (2020). This improves over work relying on zero-sum game formulation Yao (1977) where the follower's objective is assumed to be opposed to the leader's objective. An important example for the application of Stackelberg competition in trustworthy ML strategic classification. Therein, strategic individuals can, after observing the model output, adapt their data features to obtain better classification performance. Such changes in the data can cause distribution shifts that degrade the model's performance and trustworthiness on the new data, and thereby requires the companies adapt their models. In our model governance game framework, the two regulators act as *leaders* while the company acts as the *follower*. By following the Stackelberg competition, the company aims at obtaining the best-performing ML model given the requirements specified by the regulators.

# D PROOFS

**Theorem 1.** PARETOPLAY *recovers the Subgame Perfect Correlated Nash Equilibrium of* SPECGAME.

Proof via single-deviation principle from using Corollary 1 from Prokopovych & Smith (2004):

**Corollary 1** (the one-shot deviation principle for infinitely repeated games extended with an extensive form correlation device). *A pair $(Q, f)$ consisting of an extensive form correlation device $Q = \left( (M_i)_{i \in N}, \mu \right)$ and a strategy profile $f = (f_1, \ldots, f_n), f_i : H \times M_i \to \Delta(A_i)$, is a subgame perfect correlated equilibrium of $G^{\infty}(\delta)$ if and only if the one-shot deviation condition holds: no player can gain by deviating from $f$ in a single stage and conforming to $f$ thereafter.*

In the above, an extensive-form correlation device is a device that sends separately and confidentially message $M_i$ to each players $i \in N = \{\text{comp}, \text{fair}, \text{priv}\}$ at the beginning of each stage. $H$ is the history of the actions played in the prior rounds of the repeated game $G^{\infty}$.

*Proof.* In the context of SPECGAME, the correlation device $Q$ consists of messages in the form of Pareto frontiers $M_i = PF_i$ where $PF_i : \mathcal{S} \mapsto [0, 1] \times [0, 1] \times \mathbb{R}^+$ over the objectives. namely, error $err : \mathcal{S} \mapsto [0, 1]$, disparity $\Gamma : \mathcal{S} \mapsto [0, 1]$ and privacy $\mathcal{E} : \mathcal{S} \mapsto \mathbb{R}^+$ of a given $\boldsymbol{s}_i \in \mathcal{S}$ where $\mathcal{S}$ is the space of trustworthy hyper-parameters. $\mu$ is a probability distribution on the Cartesian product of these message sets $M = \underset{i \in N}{\times} M_i = \underset{i \in N}{\times} PF_i$ and the randomization is over the datasets $D_i = (X_i, Y_i) \sim \mathcal{S}$ used to tune each model. $\mathcal{D}$ is the data-generating distribution of input features $X_i \in \mathcal{X}$ and labels $Y_i \in \mathcal{Y}$.

The proof for one-shot deviation principle for SPECGAME simulated played via PARETOPLAY follows: The PARETOPLAY strategy profile $f = (f_i)_{i \in N}$ is to make gradients updates according to $PF_i$ distributed to it via the correlation device $Q$ (lines 8 and 6 in Algorithm 1).

We wish to show that no player (especially the company) can gain from deviation from $f$ in a single stage and conforming to $f$ thereafter. Assume to contrary that a player (e.g. the company) benefits from such a deviation. That is at some $t$, the company can report $\boldsymbol{s}_r$ that is not on its PF (or equivalently it performs a gradient update not following $f$). By definition, then there exists some $\boldsymbol{s}^*$ which Pareto dominates $\boldsymbol{s}_r$ : it is at least as good in all objectives and better in at least one.

We first note that reporting $\boldsymbol{s}_r$ where $err(\boldsymbol{s}_r) > err(\boldsymbol{s}^*)$ is irrational (in the game theoretic sense that it increases the agent's cost instead of reducing it) and thus never a best response for C. So we can only consider cases where it holds that either $err(\boldsymbol{s}_r) < err(\boldsymbol{s}^*)$ and $\Gamma(\boldsymbol{s}_r) > \Gamma(\boldsymbol{s}^*)$, or $err(\boldsymbol{s}_r) < err(\boldsymbol{s}^*)$ and $\mathcal{E}(\boldsymbol{s}_r) > \mathcal{E}(\boldsymbol{s}^*)$, or both hold. But every agent in Pareto Play, re-calculates its Pareto frontier as a first step (line 2 in Alg. 1). Assume, if at time $t - 1$, C adds $\boldsymbol{s}_r$ to $R^{(t)}$ .

At time $t$, the regulator would re-calculate its PF; but since $\boldsymbol{s}_r$ is not on the PF, either a) some other $\boldsymbol{s}^*$ already exists in $R^{(t)}$ which dominates $\boldsymbol{s}_r$, and therefore $\boldsymbol{s}_r$ never appears in the rest of the regulators round; or b) if no such $\boldsymbol{s}^*$ exists, the regulator will assume $\boldsymbol{s}_r$ to be a valid Pareto efficient solutions, adopt it as its initialization, and take a step on the Pareto frontier to improve the corresponding regulator loss. At this point, depending on which objective value was under-reported by C the regulator would either be able to find an $\boldsymbol{s}^*$ that Pareto dominates $\boldsymbol{s}_r$ — at which point $\boldsymbol{s}_r$ is again effectively removed from the PF calculations — or the next regulator is going to make a gradient step and find the appropriate $\boldsymbol{s}^*$ that Pareto dominates the misreported $\boldsymbol{s}_r$. In the worst-case where we lose gradient information (in a boundary condition, or near an inflection point), we note that every agent trains a model in the Calibration phase (line 9). At this point, with a near 0 gradient step, $\boldsymbol{s}^* \approx \boldsymbol{s}_r$ is re-evaluated by one of the regulators, which ensures that $\varepsilon$ and/or $\gamma$ values are corrected, which again leads to exclusion of $\boldsymbol{s}_r$ from the Pareto frontier. Therefore, the single deviation from $f$ (i.e. choosing $\boldsymbol{s}_r$ over $\boldsymbol{s}^*$ ) does not benefit the company; which is a contradiction that completes the proof. $\square$

# E FORMALIZING PRICE OF ANARCHY FOR SPECGAME

*Price of Anarchy (PoA)* is the canonical measure for quantifying this inefficiency Koutsoupias & Papadimitriou (1999). It is defined in terms of a *group cost* function $cost : \mathcal{S}^T \times \mathcal{S} \mapsto \mathbb{R}^+$ for an

outcome strategy profile $\Pi s \in \mathcal{S}^T$ and the initial specification $b \in \mathcal{S}$. The group cost combines the loss of all players into one[3]. Intuitively, PoA is the ratio of worst group cost of any equilibrium outcome in SPECGAME to the best group cost possible (as in COLLABREG). Next, we define an appropriate $cost(.)$ function for SPECGAME under PARETOPLAY.

Given a Pareto frontier $\boldsymbol{\ell}(s) = [\mathrm{err}(s) \quad \Gamma(s) \quad \mathcal{E}(s)]$ and specification $b = (\gamma^*, \varepsilon^*)$, we define the group cost of strategy $s$ for the stage game as the sum of normalized player costs:

$$q(s; b) = \frac{1}{\max_{\tilde{s}}} \{\mathrm{err}(s) + (\lambda_{fair} + C_{fair}\mathbb{1}_{[\hat{\Gamma}(s) \geq \gamma^*]})(\hat{\Gamma}(s) - \gamma^*) + (\lambda_{priv} + C_{priv}\mathbb{1}_{[\hat{\mathcal{E}}(s) \geq \varepsilon^*]})(\hat{\mathcal{E}}(s) - \varepsilon^*)\}$$

$$+ \frac{1}{\max_{\tilde{s}} \Gamma(\tilde{s})} \{\mathrm{err}(s)\mathbb{1}_{[\hat{\Gamma}(s) < \gamma^*]} + (\hat{\Gamma}(s) - \gamma^*)\mathbb{1}_{[\hat{\Gamma}(s) \geq \gamma^*]}\}$$

$$+ \frac{1}{\max_{\tilde{s}} \mathcal{E}(\tilde{s})} \{\mathrm{err}(s)\mathbb{1}_{[\hat{\mathcal{E}}(s) < \varepsilon^*]} + (\hat{\mathcal{E}}(s) - \varepsilon^*)\mathbb{1}_{[\hat{\mathcal{E}}(s) \geq \varepsilon^*]}\},$$

where the denominators are the maximum achieved error, fairness violations and privacy budget on the Pareto frontier $\boldsymbol{\ell}(s)$. Since PoA is a ratio, any valid Pareto frontier $\boldsymbol{\ell}(s)$ works for normalization provided it is used for both numerator and denominator of the PoA. See Section 3.1 for a detailed description of regulator losses.

The *group cost* for the entire Stackelberg competition under the complete strategy profile $\Pi s = \left(s^{(0)}, \ldots, s^{(t)}, \ldots, s^{(T-1)}\right)$ is $cost(\Pi s; b) = \sum_{t=0}^{T-1} c^t q(s^{(t)}; b)$, where $c$ is the discounting factor. Given a set of equilibria $Eq \subset \mathcal{S}^T$, we define the price of anarchy (PoA) as:

$$PoA_b = \frac{\max_{\Pi s \in Eq} cost(\Pi s; b)}{\min_{\tilde{s}} q(\tilde{s}; b)}. \tag{9}$$

Bounding the PoA is challenging even for simple games Koutsoupias & Papadimitriou (1999); Nisan et al. (2007). For SPECGAME, this is even more challenging given its data-dependent nature. However, we can produce an empirical PoA as a measure of equilibrium inefficiency. To do so, we estimate the Pareto frontier $\boldsymbol{\ell}(s)$ and calculate $cost(\Pi s; b)$ over the entire run of the game which produces an correlated equilibrium (see Theorem 1).

## F  REGULATOR'S INCOMPLETE INFORMATION: ESTIMATING $\lambda_{fair}$ AND $\lambda_{priv}$

The penalty scalars $\lambda_{fair}$ and $\lambda_{priv}$ are company parameters that regulators can have, at best, *incomplete information* about (Fudenberg & Tirole, 1991). Regulators using PARETOPLAY would need to estimate these parameters. In this section, we provide a systematic way to do so on a dataset they have access to.

Consider two models $\omega_1$ and $\omega_2$ that achieve the same fairness guarantee: $L_{fair}(s_1) = L_{fair}(s_2)$ **(A)**. We require that the two models achieve the same overall company loss: $\ell_{comp}(s_1) \approx \ell_{comp}(s_2)$ **(R)**.

Using Equation (4):

$$\mathrm{err}(s_2) - \mathrm{err}(s_1) = (\lambda_{priv} + C_{priv}\mathbb{1}_{[\mathcal{E}(s_1) \geq \varepsilon^*]})(\mathcal{E}(s_1) - \varepsilon^*)$$
$$- (\lambda_{priv} + C_{priv}\mathbb{1}_{[\mathcal{E}(s_2) \geq \varepsilon^*]})(\mathcal{E}(s_2) - \varepsilon^*)$$
$$= \lambda_{priv}(\mathcal{E}(s_1) - \mathcal{E}(s_2)) + C_{priv}\mathbb{1}_{[\mathcal{E}(s_1) \geq \varepsilon^*]}(\mathcal{E}(s_1) - \mathcal{E}(s_2))$$
$$= (\mathcal{E}(s_1) - \mathcal{E}(s_2))(\lambda_{priv} + C_{priv}\mathbb{1}_{[\mathcal{E}(s_1) \geq \varepsilon^*]}) \tag{10}$$

Therefore, we have:

$$\lambda_{priv} + C_{priv}\mathbb{1}_{[\mathcal{E}(s_1) \geq \varepsilon^*]} = \frac{\mathrm{err}(s_2) - \mathrm{err}(s_1)}{\mathcal{E}(s_1) - \mathcal{E}(s_2)}$$

To calibrate $\lambda_{priv}$, we want to ensure our requirement (R) is met under condition (A), so we find models in $S_\gamma = \{s \mid L_{fair}(s) = \gamma\}$ where $S_\gamma$ are the set of models that achieve fairness gap $\gamma$,

---

[3]Note that this combined measure of cost is better known as the *social cost* in the algorithmic game theory literature. We use group cost to avoid any confusion with the societal (fairness and privacy) risks.

clearly for two models $s_1$ and $s_2 \in S_\gamma$, our requirement is met. Thus, regulator's estimate $\hat{\lambda}_{priv}$ of $\lambda_{priv}$ is:

$$\hat{\lambda}_{priv} = \underset{\gamma \in [0,1]}{E} \underset{s_1, s_2 \in S_\gamma}{E} \left[ \frac{\text{err}(s_2) - \text{err}(s_1)}{\mathcal{E}(s_1) - \mathcal{E}(s_2)} - C_{priv} \mathbb{1}_{[\mathcal{E}(s_1) \geq \varepsilon^*]} \right]; \tag{11}$$

Similarly:

$$\hat{\lambda}_{fair} = \underset{\varepsilon \in [0, \varepsilon_{max}]}{E} \underset{s_1, s_2 \in S_\varepsilon}{E} \left[ \frac{\text{err}(s_2) - \text{err}(s_1)}{\Gamma(s_1) - \Gamma(s_2)} - C_{fair} \mathbb{1}_{[\Gamma(s_1) \geq \gamma^*]} \right], \tag{12}$$

where $S_\varepsilon = \{s \mid L_{priv}(s) = \varepsilon\}$ is the set of models with achieved privacy budget of $\varepsilon$.

### F.1 PARETOPLAY SETUPS

#### F.1.1 PARETOPLAY ON FAIRPATE

In FairPATE, we train teacher ensemble models on the training set. These teachers vote to label the unlabeled public data. We then train student models on the now labeled public data. At inference time, the student model does not answer all the queries in the test set. It refrains from answering a query when answering it would violate the fairness constraint. Coverage measures the percentage of queries that the student does answer.

We denote the student model for classification by $\omega$, the features as $(\mathbf{x}, z) \in \mathcal{X} \times \mathcal{Z}$ where $\mathcal{X}$ is the domain of non-sensitive attributes, $\mathcal{Z}$ is the domain of the sensitive attribute (categorical variable). The categorical class-label is denoted by $y \in [1, \ldots, K]$. We indicates the strategy vector space as $s = (\gamma, \varepsilon)$ where $\gamma$ is the maximum tolerable fairness violation and $\varepsilon$ is the privacy budget.

We train student models on a range of $s = (\gamma, \varepsilon)$ and pre-compute Pareto frontier on these results. We show Pareto frontier of UTKFace in Figure 3 and Pareto frontier of CelebA as well as FairFace in Figure 8.

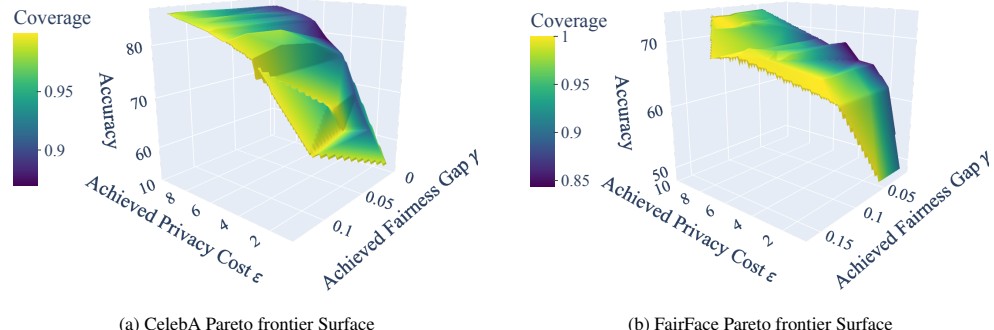

(a) CelebA Pareto frontier Surface        (b) FairFace Pareto frontier Surface

Figure 8: **Pareto frontier Surface on CelebA and FairFace**

The loss functions of all agents depend on both $\gamma$ and $\varepsilon$. A gradient descent update of $\gamma$ and $\varepsilon$ is:

$$\gamma^t = \gamma^{t-1} - \eta_{\text{fair}} \frac{\partial L}{\partial \gamma}, \ \varepsilon^t = \varepsilon^{t-1} - \eta_{\text{priv}} \frac{\partial L}{\partial \varepsilon} \tag{13}$$

The company cares about both student model accuracy and coverage. It would want to provide accurate classification and answer most queries. Its loss function uses a weighted average of the two:

$$\ell_{\text{b}}(\gamma, \varepsilon) = -(\lambda_b \text{acc}(\gamma, \varepsilon) + (1 - \lambda_b)\text{cov}(\gamma, \varepsilon)) \tag{14}$$

where $\lambda_b$ is a hyperparameter set by the company that controls how much it values accuracy and coverage. The accuracy and coverage are multiplied with -1 to form the loss because we want to maximize them. Both accuracy and coverage values used are between 0 and 1. At each turn, the

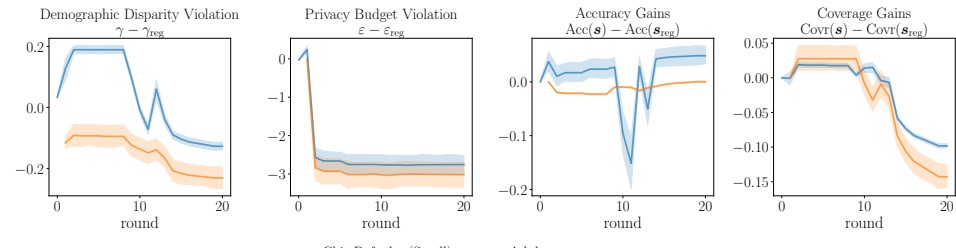

Figure 9: **Agents can have separate datasets in PARETOPLAY.** We simulate a regulator-led game where regulators have access to Adult and company has access to Chit Defaults. We observe similar trends for privacy budget and coverage metrics between the two agents; but slight differences in terms of fairness and accuracy. We attribute the differences to the feature mismatch between the two datasets (Adult and Chit Defaults) which is more prominent in tabular data (with predefined data structure) than the vision datasets (which features that are all in the pixel space) presented in the main paper.

company decides its response by calculating $\frac{\partial \ell_b}{\partial \gamma}$ and $\frac{\partial \ell_b}{\partial \varepsilon}$ at the current $\gamma$ and $\varepsilon$. In our experiments, we use $\lambda_b = 0.7$.

The loss function of the fairness and privacy regulators are $\ell_{\text{fair}}(\gamma, \varepsilon) = \gamma_{\text{ach}}(\gamma, \varepsilon)$ and $\ell_{\text{priv}}(\gamma, \varepsilon) = \varepsilon_{\text{ach}}(\gamma, \varepsilon)$ respectively.

## G  FAIRNESS

We provide more details on the fairness notions used in our empirical study in Section 4.

**Demographic Parity Fairness.**  Yaghini et al. (2023) adopt the the fairness metric of *multi-class demographic parity* which requires that ML models produce similar success rates (*i.e.*, rate of predicting a desirable outcome, such as getting a loan) for all subpopulations (Calders & Verwer, 2010).

In practice, they estimate multi-class demographic disparity for class $k$ and subgroup $z$ with: $\widehat{\Gamma}(z, k) := \frac{|\{\hat{Y}=k, Z=z\}|}{|\{Z=z\}|} - \frac{|\{\hat{Y}=k, Z\neq z\}|}{|\{Z\neq z\}|}$, where $\hat{Y} = \omega(\mathbf{x}, z)$. They define demographic *parity* when the worst-case demographic disparity between members and non-members for any subgroup, and for any class is bounded by $\gamma$:

**Definition 5** ($\gamma$-DemParity). *For predictions $Y$ with corresponding sensitive attributes $Z$ to satisfy $\gamma$-bounded demographic parity ($\gamma$-DemParity), it must be that for all $z$ in $\mathcal{Z}$ and for all $k$ in $\mathcal{K}$, the demographic disparity is at most $\gamma$: $\Gamma(z, k) \leq \gamma$.*

## H  ADDITIONAL EMPIRICAL RESULTS

**Information equality is not necessary for PARETOPLAY using tabular data.**    In Figure 5, we demonstrated that information equality is not required for PARETOPLAY using FairFace and UTKFace vision data. We repeat out experiment using tabular data in Figure 9. We simulate a regulator-led game where regulators have access to Adult and company has access to Chit Defaults. We observe similar trends for privacy budget and coverage metrics between the two agents; but slight differences in terms of fairness and accuracy. We attribute the differences to the feature mismatch between the two datasets (Adult and Chit Defaults) which is more prominent in tabular data (with predefined data structure) than the vision datasets (which features that are all in the pixel space) presented in the main paper.

**Enforcing equilibria despite incomplete information.**    Exogenous factors aside, given the uncertainty regarding the company's dataset and its parameters $\lambda_{fair}, \lambda_{priv}$ it is possible that penalties issued are not enough to avoid specification violations. If the game has converged to an undesirable equilibrium, regulators can change their penalty scalars $C_{fair}, C_{priv}$ to enforce their specification accordingly. We demonstrate this in Figure 10. The game has multiple phases in each of which we run SPECGAME until convergence. We simulate the aforementioned uncertainty by assuming no

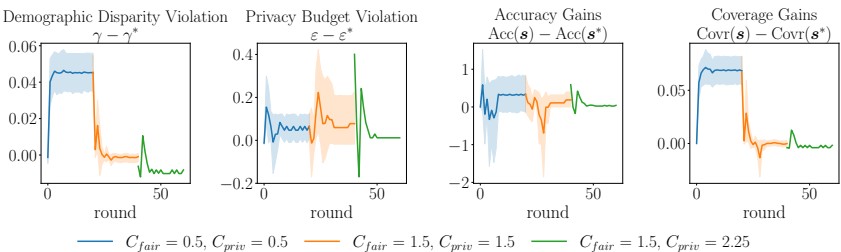

Figure 10: **Regulators can enforce desired equilibria despite incomplete information.** A scenario where initial penalties were ineffective in enforcing compliance with the specification (blue) due to incomplete information about company's loss. Regulators re-calculate their penalty scalars $C_{fair}, C_{priv}$ to progressively enforce stronger penalties in two subsequent phases of the game (orange and green) to reduce the number of violations.

priors on $C$'s, and choosing $C_{fair} = 0.5$ and $C_{priv} = 0.5$ in the first phase. As before, we simulate the outcome for 5 different initial specifications $s_{reg}$ using which we draw the 95% confidence intervals. In the first phase (blue), we observe that a large portion of the games violate disparity specifications by 6% for a similar improvement in coverage. The constraint violations are due to inappropriate penalties. In the second phase (orange) we recalculate $C_{fair} = C_{priv} = 1.5$ which manages to reduce fairness violations to 0. In this phase we get more consistent adherence with the fairness specification, but larger violations of privacy. Finally, we are left with one violation of privacy specification, increasing $C_{priv}$ to 2.25 allows us to enforce that specification as well (green).

**Utility loss due to uncertainty measured in accuracy.** Figure 4 shows uncertainty in risk estimation's impact on utility measured in both accuracy and coverage. Figure 11 shows the impact on accuracy alone. We observe that privacy has large impact on accuracy. With $\Delta\varepsilon = 1.5$, there is on average 10% reduction in accuracy on CelebA. On the other hand, fairness has negligible impact on accuracy, but more impact on coverage.

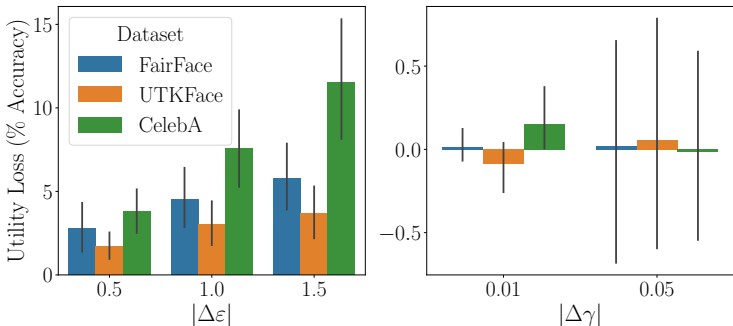

Figure 11: **Utility loss due to uncertainty in risk estimation measured in accuracy.** Uncertainty in privacy estimation has large impact on accuracy, with up to 10% reduction. Impact of uncertainty in estimation of the fairness is very small.

**Difference in Pareto frontier range can lead to constraint violation.** In Section 4 Figure 5 we showed that agents can still run PARETOPLAY when they have access to different datasets and regulators are able to enforce their constraints in most cases. However, there are scenarios where the convergence point fails to satisfy the trust constraints. We show an example of such case here in Figure 12. This example follows the same setup as in Figure 5, where regulators have access to FairFace and company has access to UTKFace. The convergence point of this game does not satisfy the fairness constraint on UTKFace but does on FairFace. This is because at the current privacy budget, higher fairness disparity gap is not achievable on FairFace. Further relaxing the fairness constraint input parameter does not lead to larger fairness gap anymore. Since regulators only have access to FairFace, they observe that the fairness gap is always below the threshold and thus do not assign any penalties. The company then continues to relax the fairness constraint input parameter without any consequences. In general, if two datasets have different ranges of fairness disparity gaps at each respective privacy budget and fairness regulators sets the fairness constraint close to the upper

| Dataset | $C_{fair}$ | $C_{priv}$ |
|---|---|---|
| UTKFace | 0.9 | 0.015 - 0.045 |
| CelebA | 1.5 | 0.1 - 0.15 |
| FairFace | 1.2 - 1.5 | 0.04 - 0.05 |
| Adult | 0.27 | 0.054 |
| Credit Card | 0.022 | 0.18 |
| Chit Defaults | 0.28 | 0.65 |

Table 4: $C_{fair}$ and $C_{priv}$ **values used in experiments.**

limit of fairness disparity gap of their dataset, the convergence point of the game may not satisfy their fairness constraint.

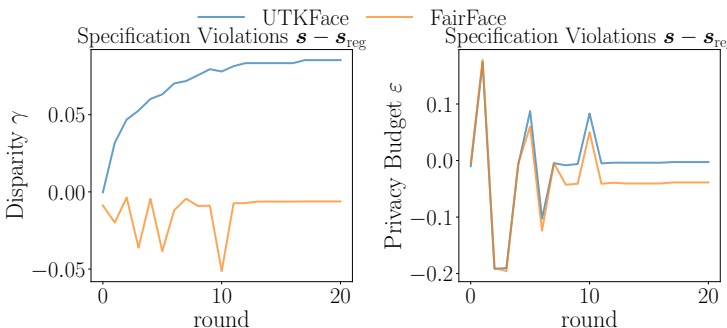

Figure 12: **Fairness regulator fails to enforce fairness constraint.** We simulate a game where agents have access to different datasets. Regulators observe that the fairness constraint is enforced on their dataset so do not assign any penalty. However, the constraint is not enforced on company's dataset and they continue to relax the fairness parameter due to lack of penalty.

## I  ADDITIONAL EXPERIMENTAL SETUP

In all games, we set step size discount factor to $c = 0.67$. For FairPATE, we use step sizes $\eta_{fair} = 0.1$ and $\eta_{priv} = 10$. We set company's internal accuracy and coverage ratio weighting to $\lambda_{\rm b} = 0.7$. See Table 4 for a list of $C_{fair}$ and $C_{priv}$ used in experiments for each dataset. We aim to use the lowest possible $C_{fair}$ and $C_{priv}$ that still enforce regulators' constraints.

The model architecture and data we use in the experiments follow what is described in the original works for FairPATE Yaghini et al. (2023). The datasets used for FairPATE and their information are shown in Table 6. For all datasets in FairPATE for the calibration step, we train the student model with Adam optimizer and binary cross entropy loss. We train for 30 epochs on UTKFace, 15 on CelebA, and 25 on FairFace.

During the games, we put box constraints on the parameters $\boldsymbol{s} = (\gamma, \varepsilon)$ so that they would not be out of range and produce undefined outputs. We use $\gamma \in [0.01, 1]$ and $\varepsilon \in [1, 10]$.

**Computational Resources.**  Experiments were conducted on a mix of 2 types of machines: (i) Machine Type I: CPU Intel Xeon Silver 4210 with 128GB RAM and GPU NVIDIA RTX 2080Ti (11GB VRAM); or (ii) Machine Type II: CPU AMD EPYC 7643 with 512GB RAM and GPU NVIDIA A100 (80GB VRAM). Game simulations without calibration run on CPU, and calibration step runs on GPU. Individual game experiments lasted 30 to 60 minutes each on vision datasets, and less than 10 minutes each on tabular dataset.

## J  LIMITATIONS

With the increasing importance of machine learning in sensitive domains, it is crucial to ensure that the machine learning models are trustworthy. However, previous research has primarily focused on

| Layer | Description |
|---|---|
| Conv2D | (3, 64, 3, 1) |
| Max Pooling | (2, 2) |
| ReLUS | |
| Conv2D | (64, 128, 3, 1) |
| Max Pooling | (2, 2) |
| ReLUS | |
| Conv2D | (128, 256, 3, 1) |
| Max Pooling | (2, 2) |
| ReLUS | |
| Conv2D | (256, 512, 3, 1) |
| Max Pooling | (2, 2) |
| ReLUS | |
| Fully Connected 1 | (14 * 14 * 512, 1024) |
| Fully Connected 2 | (1024, 256) |
| Fully Connected 2 | (256, 2) |

Table 5: Convolutional network architecture used in CelebA experiments.

| Dataset | Prediction Task | C | Sens. Attr. | SG | Total | U | Model | Number of Teachers | $T$ | $\sigma_1$ | $\sigma_2$ |
|---|---|---|---|---|---|---|---|---|---|---|---|
| CelebA | Smiling | 2 | Gender | 2 | 202 599 | 9 000 | Convolutional Network (Table 5) | 150 | 130 | 110 | 10 |
| FairFace | Gender | 2 | Race | 7 | 97 698 | 5 000 | Pretrained ResNet50 | 50 | 30 | 30 | 10 |
| UTKFace | Gender | 2 | Race | 5 | 23 705 | 1 500 | Pretrained ResNet50 | 100 | 50 | 40 | 15 |

Table 6: Datasets used for FairPATE. Abbreviations: **C**: number of classes in the main task; **SG**: number of sensitive groups; **U**: number of unlabeled samples for the student training . Summary of parameters used in training and querying the teacher models for each dataset. The pre-trained models are all pre-trained on ImageNet. We use the most recent versions from PyTorch.

addressing a single trust objective at the time or, when considering multiple objectives, assumed the existence of a central entity responsible for implementing all objectives. We highlight the limitations of this assumption for realistic scenarios with multiple agents and introduce an approach for optimization over multiple agents with multiple objectives to overcome this limitation.

Our approach recognizes the diverse nature of agents involved in deploying and auditing machine learning models. This allows us to make suggestions for guarantee levels that are more likely to be realizable in practice; given that the gains and benefits of different parties have been taken into account. We, however, acknowledge that agents may in fact have a more diverse set of requirements and objectives; and that as a result our models may not be sophisticated-enough to incorporate all such factors. Additionally, we made several assumptions regarding the economic model under which we operate as well as common knowledge of the Pareto frontier between various objectives. While these assumptions follow established principles in economics (expected utility hypothesis for the former) and in machine learning (the existence of a data-generating distribution for the latter), both are contested in their respective literature.

We acknowledge that providing "metrics" for human and society values such as fairness and privacy is imperfect at best and fraught with philosophical and ethical issues. Nevertheless, the metrics we used in our study are commonplace in trustworthy ML circles and the search for better, more inclusive, metrics is underway. Our research, therefore, aims to provide systematic guidance on best practices in regulating trustworthy ML practices, and can be adopted for future development in these areas.

From our empirical results, we observe that different ML tasks exhibit different Pareto frontiers. As such, an SPECGAME played for one task cannot necessarily provide regulation recommendation for other tasks. It remains to be seen how much such recommendations can transfer between tasks even within the same domain (for instance, vision). For instance, recommendation made on the basis of age classification may be ineffective (or too restrictive) for gender estimation.

Finally, we centered our consideration around calculating fines proportional to the privacy and fairness violations of chosen guarantee levels $(\gamma, \varepsilon)$; as well as ensuring they are effective in changing

company behavior. The converse problem is also important: assuming a bound $C$ on the penalty, what are the maximal $\gamma, \varepsilon$ guarantees that we can expect to be able to enforce?

