# OpenReview forum: "Lossgate: Incomplete Information and Misaligned Incentives Hinder Regulation of Societal Risks in Machine Learning"
_ICLR.cc/2025/Conference — Submitted to ICLR 2025_

### Official Review · Reviewer_ZndX · 2024-10-17

**Soundness:** 2
**Presentation:** 1
**Contribution:** 3
**Rating:** 3
**Confidence:** 4

**Summary:**

The paper studies how the mismatch in incentives and information between regulators and companies affects the quality of resulting machine-learning models. The authors frame this question as a principal-agent problem where the principal is the regulator, and the agent is the company. In this framework, the incentives are misaligned because the regulator is more interested in mitigating undesired social impact (e.g., fairness or privacy violations), while the company is more interested in accuracy maximization due to profit maximization incentives. At the same time, this framework also introduces information asymmetry between the regulators and the company due to the difference in evaluation datasets.

The authors analyze this problem in the following way. The end of Section 2 analyzes the committee-produced models to understand the first-best Pareto front for this problem (CollabReg). Then, the beginning of Section 3 focuses on the case where the incentives of both parties are still aligned, but the decisions are made only by the company. This analysis shows how hidden information might lead to sub-optimal decisions. Section 3.1 instantiates a specific game-theoretic setting of repeated Stackelberg games (SpecGame). Section 3.2 proposes an algorithm for finding the Nash equilibria in this setting (ParetoPlay). Finally, Section 4 empirically analyzes the equilibria produced by ParetoPlay and compares them with CollabReg equilibria. These experiments show that hidden information indeed leads to sub-optimal models in terms of the resulting accuracy and the price of anarchy. Additionally, the authors notice that SpecGame leader has a first-mover advantage, and the ParetoPlay algorithm does not result in privacy and fairness violations even in asymmetric information settings.

**Strengths:**

- The research question is relevant, given current legislative efforts such as the EU AI act, and well-motivated by the Diselgate example.
- The principal-agent framework is well-suited for the analysis. Its application for ML regulations is novel.
- The misalignment of incentives (in terms of privacy and fairness) and the asymmetry in information (in terms of different evaluation datasets) are well-contextualized in the machine-learning setting.

**Weaknesses:**

- The paper does not clearly mention the regulator's and the company's informational restrictions even though it focuses on hidden information. Sometimes, these restrictions are discernible from the context, but not everywhere. For instance, at the beginning of Section 3, the paper claims that "the vector objective $\ell(s)$ is evaluated on separate datasets," which suggests that neither the regulator nor the company knows the datasets of each other. However, Section 3.2 states that "regulators must obtain theirs (checkpoints) through a third party (e.g., public data) or the company," which suggests that the company might decide to share their dataset with the regulator and eliminate the penalty associated with hidden information.
- Similarly, even though the evaluation loss is a random variable, the paper does not clearly state which characteristic is important for participants. The analysis at the end of Section 2 suggests that the committee cares only about the empirical average of losses. However, the analysis at the beginning of Section 3 suggests that the company only cares about lower bounds for their estimates and upper bounds for the regulator's estimates.
- The incentives for the regulators in SpecGame are odd. Due to discontinuity in $\hat{s}_\omega$ variable, these incentives suggest that the regulators want the company to violate their restrictions but by a little bit because it will give them a big discontinuous advantage in $err$.

- In addition to that, the regularization penalty of the firm, which the paper interprets as fines, simply disappears in the game. However, I would expect the firm to transfer this fine to regulators, which will increase their utility.
- The actions in SpecGame are also non-standard because the regulators could directly observe and manipulate the company's action $s$.
- It is unclear which information both participants get at each step (e.g., the results of loss evaluations, the calculated Pareto frontiers, etc.).
- Line 6 in the ParetoPlay algorithm does not make sense because it adds a vector to a scalar.
- Line 3 in the ParetoPlay algorithm does not correspond to the text. As it is written now, all participants update their Pareto fronts only once.
- Calibrate procedure in the ParetoPlay algorithm is confusing. First, it is not clear whose data it is using. Second, I do not understand how exactly calibration happens.
- I have concerns about the correctness of Theorem 1. First, the analysis of the company's deviation, $s_r$, considers the case when $s_r$ does not belong to a Pareto frontier. However, the other case was not considered anywhere. Second, referenced Corollary 1 considers sequential *simultaneous* games, while the paper considers sequential *Stakelberg* games. Thus, at first glance, Corollary 1 is inapplicable to the setting of the paper.
- The algorithm for plotting Figure 4 is unclear. The parameter $\Delta \varepsilon$ is not present in SpecGame, while the algorithm for the calculation of Pareto frontiers is not presented.
- Due to the unclarity in exposition in the previous sections, the results of Section 4 are hard to interpret.

**Questions:**

- What are the informational restrictions in all settings? What do the regulators and the company observe in the SpecGame at each step?
- Which loss statistic is relevant for the regulators and the company?
- Why is the regulators' cost discontinuous? Why do the regulators not benefit from fines paid by the company?
- Why are the regulators allowed to manipulate the company's decisions but not allowed to manipulate penalties for the company in the SpecGame setting?
- Can the authors provide a mathematically correct version of Algorithm 1 and a textual description of the Calibrate procedure?
- Can the authors provide a clearer and/or more correct proof of Theorem 1 to alleviate my concerns about the correctness of this result?
- How was Figure 4 produced? By the application of Equation (5)?

---

> ### Author Response · Authors · 2024-11-28
> **Response to Weaknesses (Part 1)**
>
> First, we wish to thank the reviewer for their detailed comments and constructive feedback. We appreciate the time and effort you put into this review. While we may disagree with certain statements, we do so with all due respect for your service to the community.
>
> > The paper does not clearly mention the regulator's and the company's informational restrictions even though it focuses on hidden information. Sometimes, these restrictions are discernible from the context, but not everywhere. For instance, at the beginning of Section 3 , the paper claims that "the vector objective $\ell(s)$ is evaluated on separate datasets," which suggests that neither the regulator nor the company knows the datasets of each other. However, Section 3.2 states that "regulators must obtain theirs (checkpoints) through a third party (e.g., public data) or the company," which suggests that the company might decide to share their dataset with the regulator and eliminate the penalty associated with hidden information.
>
> The reviewer's concern regarding the comment in Section 3.2 is answered immediately  after the quoted line:
> > (Line 367) In PARETOPLAY, each agent has access to their own Pareto frontier. Companies can easily calculate Pareto frontier from their training checkpoints, but regulators must obtain theirs through a third party (e.g., public data) or the company. **In the latter case, cryptographic methods like homomorphic encryption can ensure data privacy during this process.** The specifics of regulatory data access are beyond this work’s scope but it is a crucial issue that is relevant beyond ML.
>
> That is,  the regulator can obtain checkpoints through the company itself if and only if it has access to cryptographic primitives that ensure data privacy, meaning that computation is done by the company on regulator data without the company being made aware of the regulator's dataset. Therefore, **there are no discrepancies between the two statements that the reviewer highlighted as informational restrictions do not change under homomorphic encryption.**
>
>
> > Similarly, even though the evaluation loss is a random variable, the paper does not clearly state which characteristic is important for participants. The analysis at the end of Section 2 suggests that the committee cares only about the empirical average of losses. However, the analysis at the beginning of Section 3 suggests that the company only cares about lower bounds for their estimates and upper bounds for the regulator's estimates.
>
> First, it is not true that the committee cares only about empirical average losses. The privacy and fairness loss are never average losses: the former is a worst-case loss (by definition) and the latter is a rate constraint (i.e. calculated over the full dataset, so an average loss is not meaningful).
>
> Second, there is no discrepancy between the form of the losses between Sections 2 and 3. The only difference is who gets to evaluate them (this is indicated immediately in lines 206-208)
>
> Note that in Section 3 there is no committee: therefore each player a) knows their own loss functions (thus there is no "lower bound estimation" for the company as the reviewer is suggesting), and b) estimates what they don't know.
>
>
> > The incentives for the regulators in SpecGame are odd. Due to discontinuity in $\hat{s}_\omega$ variable, these incentives suggest that the regulators want the company to violate their restrictions but by a little bit because it will give them a big discontinuous advantage in err.
>
>
> We kindly invite the reviewer to consider the penalty function in Table 1. The indicator $1_{\hat\Gamma(s) \geq \gamma}$ ensures that a penalty is applied only when there is a violation. Now if we assume that the company violates the constraint, they are inviting a penalty vs. not violating which is introducing no penalty.  **Therefore, the company is not incentivized to violate the constraint as the reviewer is suggesting.**
>
>
> > In addition to that, the regularization penalty of the firm, which the paper interprets as fines, simply disappears in the game. However, I would expect the firm to transfer this fine to regulators, which will increase their utility.
>
> The reviewer is misunderstanding the goal of a regulatory body. The regulator is not trying to gain financial benefit from penalization, it wishes to disincentivize untrustworthy model releases. Adding a gain from penalties would in turn incentivize the regulator to, for instance, apply stricter penalties (larger C values than necessary) or tighter regulation than necessary (smaller $\gamma$, $\varepsilon$ values). We acknowledge there are other rationales for modeling regulatory behavior but our model is consistent with the aforementioned goal of the regulator.

---

> ### Author Response · Authors · 2024-11-28
> **Response to Weaknesses (Part 2)**
>
> > The actions in SpecGame are also non-standard because the regulators could directly observe and manipulate the company's action $s$.
>
> Games where the players observe each others' play are not uncommon. By "observing" here, we mean that the company releases a model with trustworthy parameters $s$ that the regulator estimates. Concerning manipulation, the reviewer is not because the regulator is not directly manipulating the company's strategy, rather by choosing appropriate penalties they are forcing them to adopt s in the direction of compliance with regulation. This is also common in the context of mechanism design. Our simulator (ParetoPlay) simulates the outcome of these interactions by making the steps directly hence why there has been this confusion that agents directly manipulate each other's strategy.
>
> For better clarity, we suggest adding the following explanation:
> > The parameter $\boldsymbol s^{(t)}$ represents the trustworthy parameters that the released model at step $t$ employs. These parameters are chess pieces in the chess game between the regulator and the company. There are limited strategies that each player has (produce a model and assign penalties) that move the chess pieces in different directions. However, since the play is sequential, each round of play changes $\boldsymbol s$ from a previous state produced by the other agent's play.  This does not mean that agents are manipulating each other's actions. It simply means they are playing on the same chessboard.
>
>
> > - It is unclear which information both participants get at each step (e.g., the results of loss evaluations, the calculated Pareto frontiers, etc.).
> > - Line 6 in the ParetoPlay algorithm does not make sense because it adds a vector to a scalar.
> > - Line 3 in the ParetoPlay algorithm does not correspond to the text. As it is written now, all participants update their Pareto fronts only once.
>
>  The reviewer is correct in three instances. We have updated the pseudocode. Line 6 indicates that each regulator only updates their own parameter (hence element-wise multiplication $\odot$ ) and Line 3 estimates the Pareto frontier should not have been in the initial condition. We have also updated Line 3 to indicate the result sharing post-calibration. We thank the reviewer for their keen eye and suggestions.
>
> > Calibrate procedure in the ParetoPlay algorithm is confusing. First, it is not clear whose data it is using. Second, I do not understand how exactly calibration happens.
>
> The game recovers new trustworthy hyper-parameter values $\boldsymbol s^{(t+1)}$, the calibration simply trains a new model on the agent's dataset using $\boldsymbol s^{(t+1)}$ and reports the achieved loss values (error, privacy and fairness violations). We have expanded our discussion of the calibrate function with the following:
> > $\text{Calibrate}: \mathcal S \mapsto  [0, 1] \times [0, 1] \times \mathbb{R}^+$ is a function that takes input trustworthy parameters $\boldsymbol s \in \mathcal S$ where $\mathcal S$ is the space of trustworthy hyper-parameters, trains a model using $\boldsymbol s$ on the agent's dataset, and measures its achieved error $\operatorname{err}(\boldsymbol s)$ in $[0, 1]$, fairness violations $\Gamma(\boldsymbol s)$ in $[0, 1]$ and privacy parameter $\mathcal{E} (\boldsymbol s)$ in $\mathbb{R}^+$ and returns the tuple $(\operatorname{err}(\boldsymbol s), \Gamma(\boldsymbol s), \mathcal{E} (\boldsymbol s))$.
>
> > I have concerns about the correctness of Theorem 1. First, the analysis of the company's deviation, $s_r$, considers the case when $s_r$ does not belong to a Pareto frontier. However, the other case was not considered anywhere. Second, referenced Corollary 1 considers sequential simultaneous games, while the paper considers sequential Stakelberg games. Thus, at first glance, Corollary 1 is inapplicable to the setting of the paper.
>
> In equilibrium proofs, we seek to show that deviation from the equilibrium strategy (here $f$) does not provide a net advantage to the player deviating from $f$. Since the setting where $s_r$ is on the Pareto frontier is not a violation of the strategy profile $f$, considering it does not help us prove that $f$ is an equilibrium. We also respectfully disagree with the reviewer regarding the applicability of Corollary 1. The reason is that all extensive form games have a normal form (simultaneous-game) representation ([[Shoham and Leyton-Brown 2009]], Section 5.1.2 Page 120) while the inverse is not true. Therefore, considering this simultaneous representation, the corollary applies.

---

> ### Author Response · Authors · 2024-11-28
> **Weaknesses (Part 3) + Questions**
>
> > The algorithm for plotting Figure 4 is unclear. The parameter $\Delta \varepsilon$ is not present in SpecGame, while the algorithm for the calculation of Pareto frontiers is not presented.
>
> Figure 4 concerns Section 3 ("Hidden information leads to loss of utility for the company") before introducing *any strategic behavior,* As noted in Line 478 onwards, Figure 4 is plotted over pre-computed Pareto frontiers which are independent of SpecGame. The uncertainty parameter $\Delta \varepsilon$ is introduced in Section 3. We have added the following explanation and backward reference to Section 3 to avoid any further confusion:
> > Note that in this experiment we are not considering strategic behavior and the loss of utility is purely due to estimation uncertainty (see Section 3).
>
> > Due to the unclarity in exposition in the previous sections, the results of Section 4 are hard to interpret.
>
> Can the reviewer be more specific? What result is hard to interpret? We truly wish to ameliorate the presentation but cannot do so without objective feedback.
>
> **Questions**
>
> > What are the informational restrictions in all settings? What do the regulators and the company observe in the SpecGame at each step?
>
> They observe each other actions: release of the model and penalty assignment. As discussed earlier (with the chess board analogy), either action maps to a change in the trustworthy parameters used to train the model which means that they, equivalently,  observe the final value of $s$ post-action from the other party.
>
> > Which loss statistic is relevant for the regulators and the company?
>
> No loss statistic is irrelevant. For the company, the focus is on the error (an average loss). However, the penalty issued by the regulators also matters which is a function of the excessive privacy and fairness risks (neither are average losses—see response above). Similarly, for regulators, as long as the privacy/fairness violations are within bounds, they prefer lower error models, but once violations exceed the specification, they strictly care about these violations.
>
> > Why is the regulators' cost discontinuous? Why do the regulators not benefit from fines paid by the company?
>
> We believe we have answered this question in our response to the weaknesses.
>
> > Why are the regulators allowed to manipulate the company's decisions but not allowed to manipulate penalties for the company in the SpecGame setting?
>
> This is a misunderstanding. Regulators are not allowed to directly manipulate company decisions. Please see our extended response in the weaknesses section.
>
> > Can the authors provide a mathematically correct version of Algorithm 1 and a textual description of the Calibrate procedure?
>
> We think parts of the proof and corollary were misunderstood. We have provided clarifications. We are happy to engage in further discussion.
>
> > How was Figure 4 produced? By the application of Equation (5)?
>
> Figure 4 was produced using the Pareto frontier over $(\operatorname{err}(\boldsymbol s),  \mathcal E (\boldsymbol s) \pm |\Delta \varepsilon|, \Gamma (\boldsymbol s) \pm |\Delta \gamma|)$ achieved via training models and measuring true privacy and fairness violations $\mathcal E (\boldsymbol s)$ and $\Gamma (\boldsymbol s)$. We then plot $\operatorname{err}(\boldsymbol s)$ against $|\Delta \varepsilon|$ and $|\Delta \gamma|$ respectively by sweeping through achieved privacy/fairness violations $\mathcal E(\boldsymbol s)$ and $\Gamma (\boldsymbol s)$.
>
>
> We hope to have answered the reviewer's concerns. If so, we would appreciate it if they increased their score.

---

> ### Comment · Reviewer_ZndX · 2024-11-28
>
> Thank you for the clarifications. However, I still think that many of my concerns remain unaddressed.
>
>
> **Information restrictions.** My point about information restrictions was not about the direct discrepancies in the text but rather about the hardness of reading and associated ambiguities. I still think that the presentation of information restrictions in Section 3 can be improved. Notably, the authors should consider clearly presenting **information restrictions in Section 3.2**, where both I and Reviewer MwdX were confused about the observability of $s^{(t)}$.
>
>
> **Empirical losses.** My comment about the empirical losses was aimed at clarifying what  "each player ... estimates what they don't know" **mathematically means** in this setting. The authors' comment about the irrelevance of lower bound losses makes this task even more confusing for me because I do not have any idea how "estimates what they don't know" might enter the loss function.
>
> **Discontinuity in the loss.** I understand that the company will not be affected by this discontinuity. However, I was asking about the actions of **regulators** in my review.
>
>
> **The goal of a regulatory body.** I understand that the collection of fines is not the goal of the regulatory body. However, the **money can not simply disappear** if a company pays a fine. Please clarify who receives the fine and how this affects the participants' incentives.
>
>
> **Action spaces of participants.** If the regulators can not manipulate the action $s^{(t)}$ directly, it completely changes the analysis of the game. Thus, the authors should redo the analysis with proper action spaces and simulation.
>
>
> **Proof of Theorem 1.** First, the setting where $s_r$ is on the Pareto frontier can violate the equilibrium achieved by Algorithm 1 because not **all** points on the Partero frontier will be produced by a **single** run of gradient ascent-descent procedure. Second, Collorary 1 has very stringent requirements regarding **informational restrictions** at each step (more specifically on the observability of actions). If one naively transforms the simultaneous representation into a normal form, these restrictions will be violated, and Collorarly 1 will remain inapplicable.

---

### Official Review · Reviewer_MwdX · 2024-10-27

**Soundness:** 1
**Presentation:** 1
**Contribution:** 2
**Rating:** 3
**Confidence:** 3

**Summary:**

Much of fair and privacy-preserving machine learning framing assumes
that the developer gets to select themselves the fairness and privacy hyperparameters.
However, in practice, there is a significant tension between such social concerns
and the accuracy of the trained model.
As developers mostly profit from accuracy,
they are incentivized to weaken fairness and privacy.
The paper accounts for this,
by introducing a regulator in charge of enforcing a tradeoff between social concerns
and model accuracy.
This naturally yields a game between the developer and the regulator,
which is empirically analyzed by the paper.

**Strengths:**

By introducing the regulator,
and the tension between the machine learning developer and the regulator,
the paper makes an interesting contribution
towards making responsible AI research findings actionable for regulators.

**Weaknesses:**

Overall I found the paper poorly written,
and there are many core components that do not seem clearly laid out (as discussed below).
Moreover, I am highly unconvinced by the proof of the main theorem (Appendix D).
While this is likely due to my lack of understanding of key objects,
such as the strategy space and the costs of the two players.

In particular, at time $t$, the action seems to be a choice of $s^{(t)}$
(by the regulator at even times, and by the company at odd times).
I am not even sure whether *any* $s^{(t)}$ is a feasible action.
Now, the authors wrote costs that depend on $s$ (e.g. in Table 1).
However, it is unclear to me on which $s^{(t)}$ the value $s$ refers to.

In particular, Algorithm 1 seems to suggest that there is a single $s$
that both players can (arbitrarily?) affect at the next round.
But then, I do not understand how the regulator can affect the company,
who may simply cancel out any modification by the regulator to set their value of $s$.
Additionally, I find Theorem 1 implausible.
For one thing, Algorithm 1 involves a single gradient ascent-descent at each iteration,
which is a far cry from any kind of optimal best-response.

What is more, I do not understand why the regulator's cost is defined as a case disjunction
depending on $\hat s_w$ and $b \in \{ \gamma, \varepsilon \}$.
First, it yields a weird noncontinuity at $\hat s_w = b$,
which seems to imply that $\hat s_w = b$ is better than $\hat s_w = b - 0.00001$.
Second, it equation (2) already seemed to suggest a natural form for it.

The lack of clarity of the paper on some of its core components
prevents me from recommending acceptation.

More anecdotically:
- The 8% and 96% figures are the extreme observed values over 6 datasets.
Reporting them is misleading.
I think it would be more representative of the work to report that
"in our experiments, a few percents of utility is lost for most datasets"
and that "across six experiments, Lossgate consistently incurs around 80% increased cost".
- From a pedagogical point of view,
I would have found it useful to sketch a figure in the $(\gamma, \varepsilon)$-space,
with level lines of the company and of the regulator's losses.
In particular, this would help illustrate the ParetoPlay Gradient Descent-Ascent algorithm.

**Questions:**

Can the authors clarify the action spaces and the cost of SpecGame (especially on which $s^{(t)}$ they depend)? I am especially interested in a formal definition of the game in a classical extensive form.

Could the authors explain the rationale behind the discontinuity in the regulator's cost function at $\hat s_w = b$? How does this align with the continuous formulation in equation (2)?

There have been some recent advances in "Proofs of Training" [1].
In particular, in principle at least,
these proofs could allow the regulator to effortlessly verify
that the company's training used some claimed hyperparameters.
How would the implementation of such verification tools affect the paper's findings?

[1] Kasra Abbaszadeh, Christodoulos Pappas, Dimitrios Papadopoulos, Jonathan Katz:
Zero-Knowledge Proofs of Training for Deep Neural Networks. IACR Cryptol. ePrint Arch. 2024: 162 (2024)

---

> ### Author Response · Authors · 2024-11-28
> **Response to Weaknesses**
>
> First, we wish to thank the reviewer for their detailed comments and constructive feedback. We appreciate the time and effort you put into this review. While we may disagree with certain statements, we do so with all due respect for your service to the community.
>
>
> > I am highly unconvinced by the proof of the main theorem (Appendix D). While this is likely due to my lack of understanding of key objects, such as the strategy space and the costs of the two players. In particular, at time $t$, the action seems to be a choice of $s^{(t)}$ (by the regulator at even times, and by the company at odd times). I am not even sure whether any $s^{(t)}$ is a feasible action. Now, the authors wrote costs that depend on $s$ (e.g. in Table 1). However, it is unclear to me on which $s^{(t)}$ the value $s$ refers to.
>
> $t$ is the time index introduced in the ParetoPlay algorithm (1) in Section 3.2. In the proof we take advantage of a corollary from Prokopovych & Smith (2004) that establishes the single-deviation principle. The corollary (informally stated) argues that if we can show that one-shot deviation from a strategy profile $f$ is not beneficial then $f$ is a subgame perfect correlated equilibrium.  We consider the one-shot deviation at an arbitrary time $t$. Note that since the game is repeated, the cost structure is the same in every stage game and is equal to those tabulated in Table 1. We hope that this explanation clears the reviewer's confusion regarding the primitives we use.
>
> > Algorithm 1 seems to suggest that there is a single $s$ that both players can (arbitrarily?) affect at the next round. But then, I do not understand how the regulator can affect the company, who may simply cancel out any modification by the regulator to set their value of $s$. Additionally, I find Theorem 1 implausible. For one thing, Algorithm 1 involves a single gradient ascent-descent at each iteration, which is a far cry from any kind of optimal best-response.
>
> We note the fact that the regulator is not directly manipulating the company's strategy, rather by choosing appropriate penalties they are forcing them to adopt an $\boldsymbol s$ in the direction of compliance with the regulation. This is common in the context of mechanism design. Our simulator (ParetoPlay) simulates the outcome of these interactions by making the steps directly hence why there has been this confusion that agents directly manipulate each other's strategy—which is not the case.
>
> For better clarity, we propose to add the following explanations:
>   > The parameter $\boldsymbol s^{(t)}$ represents the trustworthy parameters that the released model at step $t$ employs. These parameters are chess pieces in the chess game between the regulator and the company. There are limited strategies that each player has (produce a model and assign penalties) that move the chess pieces in different directions. However, since the play is sequential, each round of play changes $\boldsymbol s$ from a previous state produced by the other agent's play.  This does not mean that agents are manipulating each other's actions. It simply means they are playing on the same chessboard.
>
> Regarding optimality and reviewer's claim, we note that our game is that of a Stackelberg competition which admits a particular bi-level optimization formulation. Many prior works in the algorithmic game theory literature including (Goktas & Greenwald, 2022) cited in the main paper,  show that a gradient descent ascent type algorithm (which subsumes ParetoPlay) recover the game equilibria. To see this more clearly, descent occurs in the even times whereas ascent occurs in the odd times. Note the fact that gradients of error and privacy/fairness loss are opposing each other. Therefore the comment regarding the "lack of optimality" is unsubstantiated. Theorem 1 further establishes the existence of equilibrium formally. We kindly ask the reviewer to elaborate their reasons for disregarding the proof of Theorem 1.
>
> > I do not understand why the regulator's cost is defined as a case disjunction depending on $\hat{s}_w$ and $b \in \gamma, \varepsilon$...
>
> Please see the response to your related question below.
>
> > More anecdotally:
> > The $8 \%$ and $96 \%$ figures are the extreme observed values ...
>
> We thank the reviewer for their suggestions.  We adjusted some of the noted values to be observed ranges. However, each claim made is accurate with nuances discussed in the empirical results.
>
> > From a pedagogical point of view, I would have found it useful to sketch a figure in the ( $\gamma, \varepsilon$ )-space ...
>
> We thank the reviewer for their suggestions. We note that Figures 5 (in the main paper) and 9 and 10 (in the appendix) are of the form that the reviewer seeks. However, we note that for all of these figures we have aggregated multiple runs of the algorithm to achieve error bars; hence they are plotted as a function of time (round) but always relative to the specification.

---

> ### Author Response · Authors · 2024-11-28
> **Response to Questions**
>
> > Can the authors clarify the action spaces and the cost of SpecGame (especially on which they depend)? I am especially interested in a formal definition of the game in a classical extensive form.
>
> We provide a formal definition of the repeated game in terms of its "stage game" in Line 313. The cost function and strategy space in the stage game for each player is enumerated in Table 1. Regarding the extensive form game, Figure 2 depicts the extensive form pictorially however, it should be clear that since we are dealing with a continuous strategy space the classical single branch per-available action is not applicable. Finally, we provide the formal definition of the correlation device we use (i.e., the Pareto frontiers) when presenting our notion of equilibria (Line 403). If the reviewer could provide further details on what they are seeking more than the aforementioned details, we are happy to oblige.
>
> > Could the authors explain the rationale behind the discontinuity in the regulator's cost function at ? How does this align with the continuous formulation in equation (2)?
>
> For the company, the focus is on the error (an average loss).  However, the penalty issued by the regulators also matters which is a function of the *excessive* privacy and fairness risks. Similarly, for regulators, as long as the privacy/fairness violations are within bounds, they prefer lower error models—this is because prior work has shown that weak generalization leads to excessive privacy leakage for instance, therefore in the regime where the privacy leakage is not excessive, the incentives of regulators and the company are naturally aligned. However, once violations exceed the specification, regulators strictly care about fairness/privacy violations.
>
> Regarding the reviewer's suggestion to use an equation akin to Eq. 2 for the regulators, we note that doing so implies that the regulators are always considering the company's model error as a decision factor. This is inconsistent with the aforementioned rationale as the regulator's utmost priority to avoid excessive risk to the public (who they present). We have made this rationale explicit in Line 181 when we first introduce the regulator constraints:
> > (Line 181) The two other constraints model regulators concerns about societal risk. The fairness regulator measures violations using a fairness metric $\ell_{\text {fair }}(s):=\Gamma(s) . A$ model with violation $\Gamma(s)>\gamma$ is deemed unacceptable. Similarly, the regulator measures privacy cost with $\ell_{\text {priv }}(s):=\mathcal{E}(s)$ and mandates that $\mathcal{E}(s) \leq \varepsilon$.
>
> Critically if violations are excessive, problem (1) is deemed infeasible which means that the company cannot release an untrustworthy model to the public. This is an acceptable solution to the regulator.   A loss form for the regulator akin to Eq. 2 (company loss) does not admit such a solution without complications.
>
> Finally, we point out that from a game theoretic perspective, a discontinuous loss is not uncommon. For instance, in the "grim-trigger" behavior strategy for an infinitely repeated game, if player A "defects" (does not cooperate with player B), such a strategy suggests that player B should defect indefinitely producing a discontinuous cost function.  We can consider player A as the company, player B  as the regulator, defection as releasing a model with excessive violations, and cooperation as releasing a model with violations within the specified bounds. We hope to have clarified the rationale behind our cost function and shown that discontinuities are not critical issues.
>
> > There have been some recent advances in "Proofs of Training" [1]. In particular, in principle at least, these proofs could allow the regulator to effortlessly verify that the company's training used some claimed hyperparameters. How would the implementation of such verification tools affect the paper's findings?
>
> We thank the reviewer for sharing this work. We are indeed aware of this line of work. However, we want to point out that none of these ZKP works allow "effortless verification." These often require changes to the model architecture to allow for the particular proof system and have other limitations. With regards to our work, having efficient ZKPs will simplify the game and payoff structure by removing some of the uncertainty but ultimately they do not eliminate the misaligned incentives of the players.
>
>
> We hope to have answered the reviewer's concerns. If so, we would appreciate it if they increased their score.

---

> > ### Comment · Reviewer_MwdX · 2024-11-28
> >
> > I thank the authors for their response but I am still highly unconvinced by the main (Theorem 1).
> > I stress that Goktas & Greenwald study min-max Stackelberg one-round games, where as lines 313 define a multi-stage $T$-round game.
> > I would encourage the authors to be significantly more precise in the game definition, with clearly defined action sets and information sets at each stage, along with the costs at each stage and the precise variables (with their time-dependence) the costs depend on.

---

### Official Review · Reviewer_Ce8C · 2024-10-31

**Soundness:** 2
**Presentation:** 1
**Contribution:** 2
**Rating:** 3
**Confidence:** 3

**Summary:**

This paper investigates the societal risks of machine learning (ML), such as privacy violations and bias, and explores regulatory responses aimed at mitigating these risks. It highlights the complex dynamics between companies, which prioritize model accuracy (often for profit), and regulators, who focus on minimizing societal harms. The study introduces Lossgate, a scenario where companies exploit discrepancies between their risk assessments and regulators' to maximize model accuracy. Through game theory, the authors model these regulator-company interactions in a framework they call SPECGAME, which quantifies how strategic behavior can lead to substantial societal costs—up to 96% more than if the entities collaborated. The study reveals that imperfect information and misaligned incentives between companies and regulators can either force companies to adopt overly conservative models, sacrificing accuracy, or enable them to sidestep regulation, raising societal risk.

**Strengths:**

There are a couple of merits in the paper. First, it reframes ML regulation through the principal-agent problem framework, effectively capturing the inherent misaligned incentives between regulators and companies. By modeling the relationship as a principal-agent problem, the authors illustrate how these incentives shape strategic interactions and allow for regulatory adaptation using game theory to structure and assess regulatory effectiveness in reducing societal risks. Second, the paper introduces a novel simulation algorithm specifically designed to assess the outcomes of this regulatory game. This algorithm simulates interactions between regulators and companies, allowing researchers and policymakers to identify equilibrium points within the game, or optimal outcomes under varying strategies. Through this, the paper provides a tool to validate game-theoretic predictions and explore equilibrium properties that could inform regulatory guidelines. Third, the authors emphasize the role of incomplete information in regulatory settings, noting that differences in how companies and regulators evaluate risks can lead to significant utility losses. They show that this uncertainty, especially when it comes to privacy or fairness audits, affects model outcomes and could result in over-cautious behavior from companies or, conversely, regulatory evasion strategies.

**Weaknesses:**

First, the paper claims to differentiate itself by incorporating the regulator’s perspective, yet it lacks clarity on what unique insights this approach adds. Although the game-theoretical analysis is thorough, it fails to offer any novel conclusions, as the findings align largely with expectations. Second, The a principal-agent framework is a classic model extensively studied in economics. The paper appear to be a minor adaptation of existing theories. The game-theoretic approach employed feels like a direct application rather than an innovative twist. Third, The paper attempts to integrate issues of misaligned incentives and incomplete information within a single framework. However, this combined approach appears somewhat divergent from the paper’s stated contribution of focusing on the regulator’s perspective. Moreover, the treatment of information asymmetry is underdeveloped and lacks clear distinction from data uncertainty, which may lead to conceptual overlap and confusion between these issues. This ambiguity detracts from the clarity of the framework, as readers may struggle to separate the distinct roles and impacts of asymmetric information and data uncertainty within the regulatory context. Last, the paper is not well written. Key points become difficult to track, and the narrative loses coherence, which may deter readers from fully engaging with the work. Furthermore, numerous typographical errors and inconsistencies reinforce the sense that the paper needs further refinement for readability and cohesiveness.

**Questions:**

(1) The author should polish the writing and clearly establish the paper’s main research question at the outset. The specific problem that solving the SPECGAME framework addresses should be well-articulated early on to guide the reader.

(2) The paper should explicitly outline its contributions. It would be beneficial to specify whether the paper’s primary contribution is the new insights derived from the game-theoretic model or whether it offers a practical, implementable approach. This distinction will clarify the paper's intended impact and relevance.

(3) To enhance clarity, the results should be contextualized. For instance, when discussing the privacy budget, the statement that it is “on average 6 lower” is vague. Providing a comparative benchmark or explaining the significance of this value would make the results more interpretable.

(4) Since the paper examines both fairness and privacy risks, it would strengthen the argument to discuss the interaction between these two risks in this framework. A comparative analysis of results when each risk is considered in isolation versus jointly could provide valuable insights into the model’s added value in addressing multiple societal risks simultaneously.

(5) The part of incomplete information is confusing. Without the asymmetric information, there is still data uncertainty issue. It does not seem that the paper could disentangle these two mechanisms. Alternative, the paper should discuss what the conclusion be without the issue of incomplete information, and then discuss what’s different with the presence of incomplete information.

(6) To highlight the model’s practical relevance, the numerical analysis should include a comparison between the game-theoretic approach and existing methodologies. This would provide potential users, especially applied audiences, with a clear view of the added benefits or unique insights offered by the proposed framework.

(7) In the data analysis, it’ll be helpful if you could explain how the confidence intervals are obtained. It seems they are based on 5 simulations may raise questions about statistical robustness.

(8) The paper assumes a simple linear combination when integrating three types of losses into the framework. This seems too simple to be true. It might be helpful to have some justifications.

(9) In equation (2), the dimensions of the cost function and trustworthy specification bounds appear inconsistent.

(10) The writing is rough with many typos. For instance, line 176 equ(1), subscript “acc” should be “comp”; line 343, two equations (3) and (3), etc.

---

> ### Author Response · Authors · 2024-11-28
> **Response to Weaknesses**
>
> First, we wish to thank the reviewer for their detailed comments and constructive feedback. We appreciate the time and effort you put into this review. While we may disagree with certain statements, we do so with all due respect for your service to the community.
>
>
> > First, the paper claims to differentiate itself by incorporating the regulator’s perspective, yet it lacks clarity on what unique insights this approach adds. Although the game-theoretical analysis is thorough, it fails to offer any novel conclusions, as the findings align largely with expectations.
>
> We are unsure what constitutes a "novel" conclusion from the perspective of the reviewer. Currently, regulations are neither data-driven nor task-adapted. Our framework is the first to address both issues consistently in the context of trustworthy ML. Notably, we have highlighted the direct benefits of simulating regulator-company behavior to the regulator using the novel concept of ***virtual* regulatory sandbox**:
> > (Line 093) Simulating the outcomes of SPECGAME benefits both regulators and companies. For companies, we show that even in the absence of strategic behavior, imperfect information leads to excessive utility loss—by up to 8%. This is the result of uncertainty in privacy risk estimations. Hence, increased transparency from the company can in fact benefit the company itself. For regulators, our work stresses the need for regulation that is not only data-driven (Hildebrandt, 2018) and task-adapted (Coglianese, 2023) but also cognizant of the socioeconomic context of ML models. SPECGAME enables this because regulators can simulate the outcome of their policies in a virtual regulatory sandbox (Jeník & Duff, 2020) before deploying them.
>
> Our results, although may be "expected" by a macro-economist, are concretized from the perspective of algorithmic fairness and privacy regulation for the first time. Therefore, we find the reviewer statements ignore key contributions of our work by minimizing it to a specific field of study and ignoring its multidisciplinary nature.
>
> > Second, The principal-agent framework is a classic model extensively studied in economics. The paper appear to be a minor adaptation of existing theories. The game-theoretic approach employed feels like a direct application rather than an innovative twist.
>
> We have neither claimed to have contributed to the literature of Principle-Agent Problems in economics; nor is ICLR an economics venue. We cast our problem as a PAP because the context of our work is ML regulation, and prior attempts at industrial regulation (for instance, environmental regulation) have benefited from this modeling. As with the prior comment, the reviewer is seeking a subjective "innovative twist" in our PAP modeling. Our innovation is the modeling of the ML regulation/company relationship presented in Sections 3 and 3.1. Can the reviewer provide examples of privacy and fairness regulation modeled as a PAP in the economics literature that we are unaware of?
>
> > Third, The paper attempts to integrate issues of misaligned incentives and incomplete information within a single framework. However, this combined approach appears somewhat divergent from the paper’s stated contribution of focusing on the regulator’s perspective.
>
> We cannot follow the reviewer's line of argument here. First, misaligned incentives and incomplete information are two facets of any Principle-Agent problem: given the separation of agents, they can have misaligned incentives and they can have incomplete information regarding each others' data and actions. Second, we are not solely focusing on the regulator's perspective. The introduction to Section 3 (before 3.1) is completely dedicated to the adverse effect of uncertainty on company utility. As a matter of principle, one cannot ignore any agent's perspective in game modeling.
>
> Our attention to regulatory policy is due to our application scenario (ML Regulation), which is in no way diminutive of the issue of Regulation Compliance which is regulator-facing and addressed in sufficient detail in the intro to Section 3.
>
> > Moreover, the treatment of information asymmetry is underdeveloped and lacks clear distinction from data uncertainty, which may lead to conceptual overlap and confusion between these issues...
>
> We have already acknowledged that incomplete information can take different forms (Lines 206-208), and specifically singled out data inequality as a prototypical example. The reasons for this are, a) our audience is ML-oriented and therefore quite familiar with the distribution shift phenomenon, and b) we have empirical results on impact data inequality in Sections 4 and H. We use "uncertainty" exclusively to highlight the uncertainty of trustworthy guarantee estimation in Section 3—consistent with trustworthy ML literature.  Perhaps the reviewer has construed "data uncertainty"  as used in another literature (perhaps economics?) causing the confusion.

---

> ### Author Response · Authors · 2024-11-28
> **Response to Questions (Part 1)**
>
> > (2) The paper should explicitly outline its contributions. It would be beneficial to specify whether the paper’s primary contribution is the new insights derived from the game-theoretic model or whether it offers a practical, implementable approach. This distinction will clarify the paper's intended impact and relevance.
>
> We have a clear list of contributions at the end of the introduction. We are curious to know how that list differs from the reviewer's expectations
>
> > (3) To enhance clarity, the results should be contextualized. For instance, when discussing the privacy budget, the statement that it is “on average 6 lower” is vague. Providing a comparative benchmark or explaining the significance of this value would make the results more interpretable.
>
> Talking about the privacy parameter $\varepsilon$ is inherently tricky due to the fact that it is an upper bound on the privacy loss which itself is a ratio calculated under a supremum. This makes statements such as a privacy budget increase of 10% etc. misleading (since for example, a 10% increase from 1 to 1.1 is much different than an increase from 8 to 8.8). Privacy engineers and practitioners on the other hand are quite familiar with  "the privacy parameter $\varepsilon$" and its typical values in different applications hence our choice for reporting. We always make sure to state that we report "privacy parameter $\varepsilon$" to avoid vagueness in this regard.
>
> (4) Since the paper examines both fairness and privacy risks, it would strengthen the argument to discuss the interaction between these two risks in this framework. A comparative analysis of results when each risk is considered in isolation versus jointly could provide valuable insights into the model’s added value in addressing multiple societal risks simultaneously.
>
> There are prior works that already discuss privacy/fairness interactions in much detail (under central non-strategic settings) ([[Esipova et al. 2022]], [[Tran et al 2021]], [[Yaghini et al. 2023]], etc.). A game-theoretic treatment of such interaction would mean that different regulators are competing with each other, which is an interesting idea. However, given that most regulatory bodies today operate under the jurisdication of a single government which can negotiate a trade-off between these concerns, modeling such a competition would be an academic exercise that is presently outside of the scope of our work.
>
> (5) The part of incomplete information is confusing. Without the asymmetric information, there is still data uncertainty issue. It does not seem that the paper could disentangle these two mechanisms. Alternatively, the paper should discuss what the conclusion be without the issue of incomplete information, and then discuss what’s different with the presence of incomplete information.
>
> The reviewer is correct that there will always be "data uncertainty" in estimation. However, note that in the setting where the regulator and company are collaborating they *share* data and make a collective decision. Therefore, once they both agree on an estimated privacy and fairness risk and have bounded it then the company would not be penalized. In other words, we care about the *excessive* "data uncertainty". As a result of the separation of agents, the company does not know the data that the regulator employs for privacy/fairness auditing. This creates the aforementioned excessive data uncertainty beyond that of just general data uncertainty.
>
> (6) To highlight the model’s practical relevance, the numerical analysis should include a comparison between the game-theoretic approach and existing methodologies. This would provide potential users, especially applied audiences, with a clear view of the added benefits or unique insights offered by the proposed framework.
>
> Our estimation price of anarchy is exactly presented to address this comment. It measures the ratio of the worst possible outcome in the strategic setting to the best possible outcome in the collaborative setting. What is measured is the sum cost of all parties (normalized to avoid favoring one agent over another). We present this at the end of Section 3 (Line 425) and provide empirical measurements in Table 2. Our results show that under strategic behavior there is a potential 70 to 96% increase in the sum cost of all agents which shows that if regulators and the company can find a way to collaborate they would both benefit substantially by eliminating the prospective of harmful strategic behavior.
>
> (7) In the data analysis, it’ll be helpful if you could explain how the confidence intervals are obtained. It seems they are based on 5 simulations may raise questions about statistical robustness.
>
> We have consistently mentioned that our confidence intervals are 95% taken over 5 simulations. Our experiments require re-training of a large number of models hence the limited runs for each. We are happy to oblige if this addresses a key concern of the reviewer.

---

> ### Author Response · Authors · 2024-11-28
> **Response to Questions (Part 2)**
>
> > (8) The paper assumes a simple linear combination when integrating three types of losses into the framework. This seems too simple to be true. It might be helpful to have some justifications.
>
> In Section 3.1 (Line 324) we methodically derive our loss functions using the proportionality principle. Furthermore, in the same section, we show that our loss functions admit a new interpretation of distributed (i.e., multi-party) optimization of the CollabReg setting. These form two strong justifications for our loss functions that have eluded the reviewer.
>
> > (9) In equation (2), the dimensions of the cost function and trustworthy specification bounds appear inconsistent.
>
> The reviewer is correct. We have amended the issue. We note that in Line 179-181 we explain that thereon we omit the explicit constraint on $\ell_acc$, we reflect this in the penalty scalar 0 in $\boldsymbol c$ and a bound 1 in the bounds vector $\boldsymbol b$.
>
> > (10) The writing is rough with many typos. For instance, line 176 equ(1), subscript “acc” should be “comp”; line 343, two equations (3) and (3), etc.
>
> We thank the reviewer for their keen eye. We have fixed these issues in the updated text.
>
>
> We hope to have answered the majority of the reviewer's concerns. If so, we would appreciate it if they increased their score.

---

### Author Response · Authors · 2024-11-28
**General Response**

We thank the reviewers wholeheartedly for their detailed comments and constructive feedback.

We are happy to find that all reviewers found our research question relevant and timely especially given "current legislative efforts such as the EU AI act", and thought that our setting was "well-motivated by the Diselgate example."

We are glad that reviewers found that our framework "reframes ML regulation through the principal-agent problem framework, **effectively capturing the inherent misaligned incentives between regulators and companies**"

In terms of immediate applications, a reviewer noted that "the paper provides **a tool to validate game-theoretic predictions and explore equilibrium properties that could inform regulatory guidelines**"

In terms of impact, a reviewer noted that "**the paper makes an interesting contribution towards making responsible AI research findings actionable for regulators.**"

In terms of contribution, a critical reviewer found that "By introducing the regulator, and the tension between the machine learning developer and the regulator, **the paper makes an interesting contribution towards making responsible AI research findings actionable for regulators.**

**New Manuscript and Color Codings** We have applied the reviewers' comments within the manuscript. We have highlighted newly added parts in "blue" and fixed typos in "red". As it stands the manuscript is 1 paragraph over the page limit due to the requested changes. We are confident we can regain that space, however, to avoid making additional changes we have avoided shortening the text.

We believe we have addressed most of the reviewers' comments. We are happy to engage further with reviewers and hope that the reviewers consider raising their scores.

---

### Meta-Review · Area_Chair_WMSy · 2024-12-07

**Metareview:**

**(a) Summary of Scientific Claims and Findings:**
The paper introduces **Lossgate**, a framework analyzing regulator-company interactions in machine learning, where misaligned incentives and incomplete information hinder effective regulation of societal risks like fairness and privacy violations. Using **SpecGame**, a game-theoretic model, the paper shows strategic behavior by companies can increase societal costs by 70–96% compared to collaborative approaches. The paper highlights challenges with regulatory enforcement and proposes a **ParetoPlay** algorithm to explore equilibria in such settings.

---

**(b) Strengths:**
1. **Timeliness:** Relevant to ongoing legislative efforts, such as the EU AI Act.
2. **Novel Contextualization:** Innovative application of principal-agent framework to ML regulation.
3. **Impactful Framework:** Provides tools for regulators to simulate outcomes and craft policies.
4. **Insightful Results:** Quantifies the cost of strategic behavior, emphasizing the importance of collaboration.

---

**(c) Weaknesses:**
1. **Presentation:** Writing lacks clarity; many sections are hard to follow.
2. **Conceptual Ambiguities:** Discrepancies in defining informational restrictions and unclear rationale for discontinuities in the regulator's cost function.
3. **Theoretical Concerns:** Validity of **Theorem 1** questioned due to inconsistencies in game definitions and application of corollaries.
4. **Algorithmic Clarity:** ParetoPlay and calibration steps are insufficiently detailed, leading to confusion about implementation.

---

**(d) Decision and Reasons for Rejection:**
The paper is **rejected** because:
1. Reviewers found that the core concepts (e.g., information restrictions, regulator actions) are ambiguously defined.
2. Empirical results are inadequately justified and poorly contextualized.
3. Key theoretical claims, including **Theorem 1**, lack sufficient rigor and clarity.
4. Presentation issues hinder accessibility and impact, making the work incomplete for publication at this stage especially to the ICLR audience.

**Additional Comments On Reviewer Discussion:**

The rebuttal period addressed concerns about the clarity of the framework, the soundness of theoretical proofs (notably Theorem 1), and the robustness of empirical analysis. Reviewers highlighted ambiguities in action spaces, information restrictions, and calibration procedures, with concerns about discontinuities in regulatory costs and the applicability of referenced corollaries. The authors clarified key aspects, corrected algorithmic errors, and expanded on the novelty of their interdisciplinary approach to ML regulation. However, persistent issues with clarity, incomplete proofs, and empirical robustness outweighed the strengths, leading to a decision to reject. I encourage the authors to rewrite this paper with general ICLR audience in mind and resubmit the paper.

---

### Decision · Program_Chairs · 2025-01-22

Reject